# Multiple ageing effects on testicular/epididymal germ cells lead to decreased male fertility in mice

Tsutomu Endo[1,2,3,4 ✉], Kiyonori Kobayashi[2,5], Takafumi Matsumura[2,6], Chihiro Emori [2], Manabu Ozawa[7], Shimpei Kawamoto [2], Daisuke Okuzaki [2], Keisuke Shimada [2], Haruhiko Miyata [2], Kentaro Shimada[2,6], Mayo Kodani[2,6], Yu Ishikawa-Yamauchi [7,8], Daisuke Motooka[2], Eiji Hara [1,2,5,9] & Masahito Ikawa [1,2,6,7,9 ✉]

In mammals, females undergo reproductive cessation with age, whereas male fertility gradually declines but persists almost throughout life. However, the detailed effects of ageing on germ cells during and after spermatogenesis, in the testis and epididymis, respectively, remain unclear. Here we comprehensively examined the in vivo male fertility and the overall organization of the testis and epididymis with age, focusing on spermatogenesis, and sperm function and fertility, in mice. We first found that in vivo male fertility decreased with age, which is independent of mating behaviors and testosterone levels. Second, overall sperm production in aged testes was decreased; about 20% of seminiferous tubules showed abnormalities such as germ cell depletion, sperm release failure, and perturbed germ cell associations, and the remaining 80% of tubules contained lower number of germ cells because of decreased proliferation of spermatogonia. Further, the spermatozoa in aged epididymides exhibited decreased total cell numbers, abnormal morphology/structure, decreased motility, and DNA damage, resulting in low fertilizing and developmental rates. We conclude that these multiple ageing effects on germ cells lead to decreased in vivo male fertility. Our present findings are useful to better understand the basic mechanism behind the ageing effect on male fertility in mammals including humans.

[1] Immunology Frontier Research Center, Osaka University, Osaka, Japan. [2] Research Institute for Microbial Diseases, Osaka University, Osaka, Japan. [3] Department of Experimental Animal Model for Human Disease, Center for Experimental Animals, Tokyo Medical and Dental University, Tokyo, Japan. [4] Graduate School of Agricultural and Life Sciences, The University of Tokyo, Tokyo, Japan. [5] Graduate School of Frontier Biosciences, Osaka University, Osaka, Japan. [6] Graduate School of Pharmaceutical Sciences, Osaka University, Osaka, Japan. [7] The Institute of Medical Science, The University of Tokyo, Tokyo, Japan. [8] Department of Regenerative Medicine, Yokohama City University Graduate School of Medicine, Kanagawa, Japan. [9] Graduate School of Medicine, Osaka University, Osaka, Japan. ✉email: atendo@g.ecc.u-tokyo.ac.jp; ikawa@biken.osaka-u.ac.jp

Male fertility requires sufficient numbers of functional spermatozoa that can contribute to embryonic development. Reproductive cessation occurs in aged females, whereas male fertility gradually declines but persists almost throughout life[1–3]. We used the mouse as a model to understand the effect of ageing on germ cells in the testis and epididymis: two pivotal male reproductive organs in which spermatozoa are produced, matured, and stored until ejaculation.

Within the adult mammalian testis, spermatogenesis (the program of sperm production) is carefully regulated, ensuring that large numbers of spermatozoa are produced at a constant rate (Supplementary Fig. 1a). In mice, spermatogenesis begins with undifferentiated type A spermatogonia, which encompass the spermatogonial stem cells (SSCs)[4–7]. The undifferentiated spermatogonia periodically undergo spermatogonial differentiation to become differentiating spermatogonia (also known as $A_1/A_2/A_3/A_4$/intermediate/B spermatogonia). Germ cells then become spermatocytes and undergo meiotic initiation[8,9]. This begins with DNA replication followed by two cell divisions, resulting in the formation of haploid, round spermatids, which develop into spermatozoa. All germ cells are located in the seminiferous tubule of the testis and supported by somatic (Sertoli) cells, which supply various factors essential for spermatogenesis[10]. Spermatozoa are released into the lumen of the seminiferous tubules (called spermiation) and acquire their motility and fertility as they migrate through the head (caput), body (corpus), and distal (cauda) epididymis[11,12]. Finally, these functional spermatozoa are stored in the cauda epididymis until ejaculation (Supplementary Fig. 1a).

The undifferentiated spermatogonia have a remarkable capacity for self-renewal and differentiation: they can reconstitute spermatogenesis upon transplantation to a germ cell-depleted testis[13,14]. The effects of ageing on SSCs are well documented in vitro and in vivo; the transplantation technique has revealed that SSC numbers and activity decline by 24 months of age and are affected by somatic environment in the aged testis[15,16]. In vitro, the SSCs derived from neonatal mice can produce offspring via transplantation up to 24 months of culture ($10^{85}$-fold expansion), but no spermatozoa are developed after 30 months ($10^{105}$ to $10^{106}$-fold expansion)[17,18]. By contrast, the in vitro SSCs derived from adult mice at 3–4 months of age undergo a decline of their potential to undergo spermatogenesis after 4–6 months of continuous expansion culture[19]. However, the detailed effects of ageing on germ cells during and after spermatogenesis, in aged testicular and epididymal environments, respectively, have remained unclear. Moreover, although several studies collectively exhibit the impact of ageing on male fertility, sperm parameters, and testis histology[20], it is difficult to interpret their correlations because each piece of evidence was provided by different species/strains, time point settings, and experimental conditions. To address these questions, we first investigated in vivo male fertility with age, using C57BL/6 J mice as a model. We then comprehensively explored the overall organization of the aged testis and epididymis, focusing on spermatogenesis, and sperm function and fertility. We conclude that multiple ageing effects on germ cells during and after spermatogenesis lead to decreased in vivo male fertility.

## Results

### In vivo male fertility decreases with age.

We monitored in vivo male fertility in C57BL/6 J mice by mating test from 2 to 24 months of age (Fig. 1a, b), when the mice have shown ageing signs such as graying hair and body weight loss (from 18 to 24 months) (Fig. 2c). The number of pups per litter (litter size) derived from females gradually decreased until 24 months of age ($P = 0.0003$, Pearson's correlation) (Fig. 1c), consistent with a previous report[21]. The litter size was significantly low at 22–24 months ($P = 0.004$, Dunnett's test) (Fig. 1c), and pup numbers per vaginal plug in females decreased (albeit not significantly) with male age (Supplementary Fig. 1b). Pregnancy rates in females exhibiting vaginal plugs (litters per vaginal plug) were not affected with male age (Supplementary Fig. 1c), indicating that the reduced litter size (Fig. 1c) is a main symptom of decreased in vivo fertility in aged males.

We then tested for the intervals between successive births (litters) in females after mating (Fig. 1d). Days between births (and days between birth and the next pregnancy) in females were not significantly extended with male age (Fig. 1e and Supplementary Fig. 1d), suggesting that frequency of mating behavior did not decrease with male age. Because male sexual behavior is associated with testosterone[22,23], we predicted that testosterone levels in aged males should be normal. Indeed, both serum and testicular testosterone concentrations in 24-month-old males were not decreased (Fig. 1f, g). We conclude that in vivo male fertility decreases with age regardless of their mating behaviors or testosterone levels.

### Aged testes show impaired sperm production.

Because the decreased fertility in aged males was not due to the decreased mating frequency, we hypothesized that the testis and/or epididymis should undergo functional decline. To test this prediction, we first focused on changes in testes with age. Testis sizes and weights were similar between 2 and 24 months of age (Fig. 2a, b), whereas testis weights per body weight were significantly decreased with age (Fig. 2c, d).

We then compared the testis histologies at 2 and 24 months. In any given seminiferous tubule cross-section, one sees stereotypical associations of germ cells at different steps of their development into spermatozoa[24]. In mice, these associations can be classified into 12 patterns, known as seminiferous stages I–XII[25]. At 2 months, almost all tubules contained multiple layers of germ cell types, whereas at 24 months 8.4% of tubules showed germ cell depletion (Fig. 2e, f and Supplementary Fig. 2a, b). We then observed their germ cell associations. In the testis, spermatozoa are released into the lumen of seminiferous tubules in stage VIII (before stages IX-X)[11,26]. At 24 months, the number of stage IX-X tubules with aligned spermatozoa, which were still located along the luminal edge of the tubules, per total tubules was increased to 5.8%, vs. 1.7% at 2 months (Fig. 2g, h and Supplementary Fig. 2a, c). Similarly, the number of stage IX-X tubules with aligned spermatozoa per total stage IX-X tubules was increased to 42.4% at 24 months vs. 10.1% at 2 months (Fig. 2i), indicating that nearly half of aged tubules exhibit sperm release failure when spermatozoa reach stage VIII. Moreover, at 24 months, 6.3% of tubules contained more than two stages in a tubule cross-section area, which were not observed at 2 months (Fig. 2j, k and Supplementary Fig. 2a). We conclude that about 20% of seminiferous tubules show abnormal stages, including germ cell depletion, in aged testes.

The remaining 80% of seminiferous tubules showed normal germ cell association patterns at 24 months (Supplementary Fig. 2a). However, these aged tubules contained thin layers of germ cells (Fig. 3a and Supplementary Figs. 2a and 3a). We thus counted germ cells of these aged tubules in stage VII, when multiple types of germ cells are ready to undergo key transitions of spermatogenesis: spermatogonial differentiation, meiotic initiation, spermatid elongation, and sperm release[26,27]. Indeed, the number of type A spermatogonia, which are at undifferentiated states in stage VII[28–30], was significantly decreased at 24 months (Fig. 3a, b and Supplementary Fig. 3a). Because the

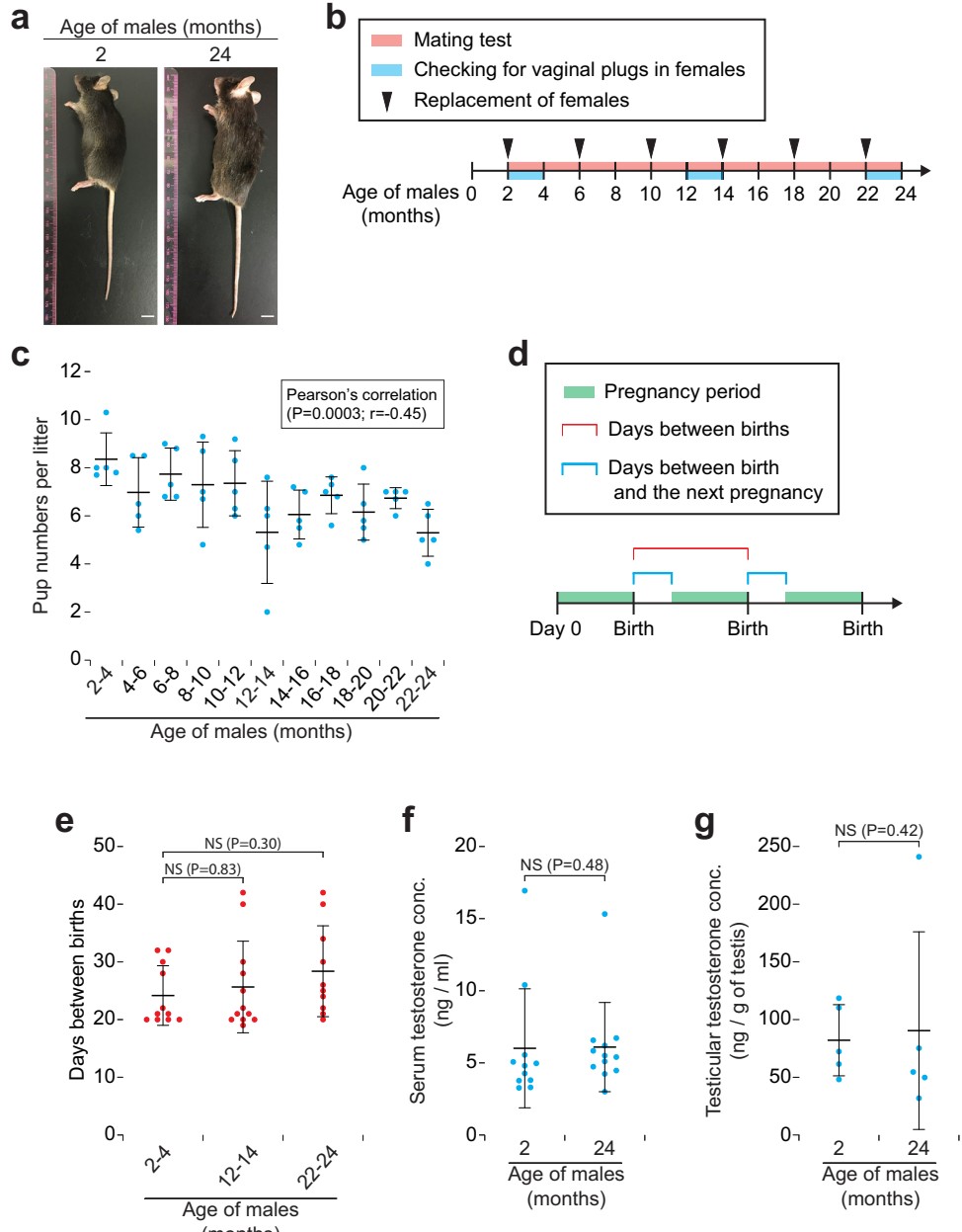

**Fig. 1 In vivo fertility decreases with age in male mice. a** Appearance of C57BL/6 J male mice at 2 and 24 months of age. Scale bars, 10 mm. **b** Diagram of mating test. Single 2-month-old males ($n = 5$) were caged with two of 2-month-old females, and the females were replaced with new females every 4 months (arrowheads), until the males reached the age of 24 months (red box). Vaginal plugs in females were checked when the males reached the age of 2–4, 12–14, and 22–24 months. **c** Mating test (pup numbers per litter) with males from 2 to 24 months of age. Error bars, mean ± SD. Blue dots, biological replicates of males ($n = 5$). $P = 0.0003$ and $r = -0.45$ (Pearson's correlation). **d** Diagram of days between birth and the next pregnancy. Green box, pregnancy period. Red bar, days between litters (births). Blue bars, days between birth and the next pregnancy. **e** Days between births. Each red dot represents the days between births when females ($n = 30$) were mated with males ($n = 5$) at 2–4, 12–14, or 22–24 months of age. NS, not significant ($P > 0.05$; Dunnett's test). **f** Serum testosterone concentration (ng/mL) in males at 2 and 24 months of age. Blue dots, biological replicates of males at 2 ($n = 11$) and 24 ($n = 12$) months of age. NS, not significant ($P > 0.05$; one-tailed $t$ test). **g** Testicular testosterone concentration (ng/g of testis) in males at 2 and 24 months of age. Blue dots, biological replicates of males at 2 ($n = 5$) and 24 ($n = 5$) months of age. NS, not significant ($P > 0.05$; one-tailed $t$ test).

number of undifferentiated spermatogonia specifies the total number of their descendant germ cells, the numbers of spermatocytes, spermatids, and spermatozoa should correspondingly decrease with age. As predicted, these cell numbers were decreased at 24 months (Fig. 3c). We conclude that the efficiency of sperm production is impaired in whole-aged testes.

**Sperm numbers in epididymides are decreased with age.** We next focused on changes in epididymides with age. Both epididymis sizes and weights were increased with age (Fig. 3d, e); epididymis weights per body weight were stable with age (Fig. 3f), indicating that the changes in epididymis weights correspond with body growth. Because sperm production, including sperm release (Fig. 2g–i and Supplementary Fig. 2a, c), was impaired in aged

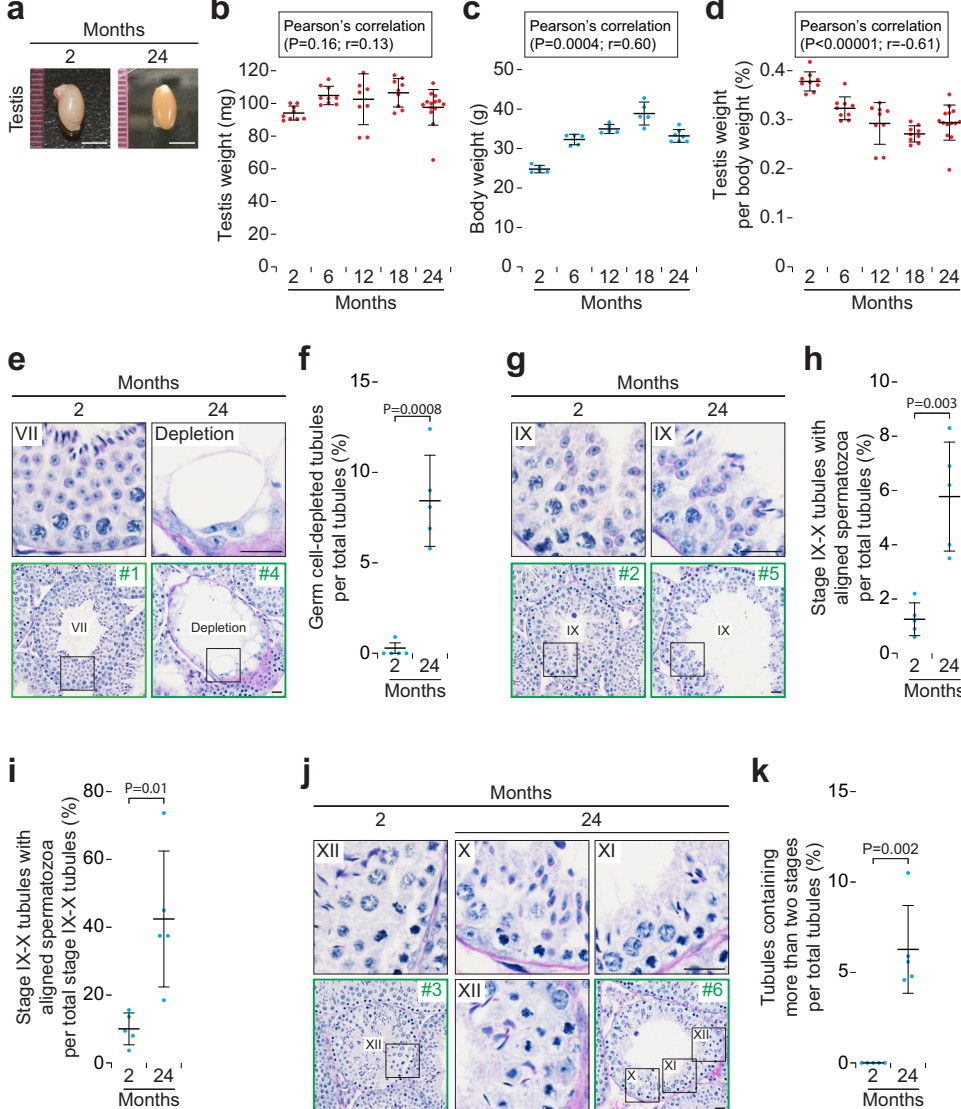

**Fig. 2 Aged testes show partial depletion of germ cells and abnormal germ cell associations. a** Gross morphology of testes in males at 2 and 24 months of age. Scale bars, 5 mm. **b** Testis weight (mg) in males at 2, 6, 12, 18, and 24 months of age. Error bars, mean ± SD. Red dots, testis values in males at 2 ($n = 5$), 6 ($n = 5$), 12 ($n = 5$), 18 ($n = 5$), and 24 ($n = 7$) months of age. $P = 0.16$ and $r = 0.13$ (Pearson's correlation). **c** Body weights (g) in males at 2, 6, 12, 18, and 24 months of age. Error bars, mean ± SD. Blue dots, biological replicates of males at 2 ($n = 5$), 6 ($n = 5$), 12 ($n = 5$), 18 ($n = 5$), and 24 ($n = 7$) months of age. $P = 0.0004$ and $r = 0.60$ (Pearson's correlation). **d** Testis weight per body weights (%) in males at 2, 6, 12, 18, and 24 months of age. Error bars, mean ± SD. Red dots, testis values in males at 2 ($n = 5$), 6 ($n = 5$), 12 ($n = 5$), 18 ($n = 5$), and 24 ($n = 7$) months of age. $P < 0.00001$ and $r = -0.61$ (Pearson's correlation). **e** Testis tubule cross-sections in males at 2 (*Left*; a stage VII tubule) and 24 (*Right*; a tubule showing germ cell depletion) months of age, stained with hematoxylin and periodic acid-Schiff (He-PAS). *Upper* panels enlarge black boxed regions in *Lower* panels. Green boxed regions (#1 and #4) indicate areas shown in higher magnification in Supplementary Fig. S2a. Scale bars, 20 μm. **f** Number of germ cell-depleted tubule cross-sections per total tubule cross-sections. Blue dots, biological replicates of males at 2 ($n = 5$) and 24 ($n = 5$) months of age. $P = 0.0008$ (one-tailed $t$ test). **g** Testis tubule cross-sections in stage IX of males at 2 (Left) and 24 (Right) months of age, stained with He-PAS. *Upper* panels enlarge black boxed regions in *Lower* panels. Green boxed regions (#2 and #5) indicate areas shown in higher magnification in Supplementary Fig. S2a. Scale bars, 20 μm. **h** Number of stage IX-X tubule cross-sections with aligned spermatozoa per total tubule cross-sections. Blue dots, biological replicates of males at 2 ($n = 5$) and 24 ($n = 5$) months of age. $P = 0.003$ (one-tailed $t$ test). **i** Number of stage IX-X tubule cross-sections with aligned spermatozoa per total stage IX-X tubule cross-sections. Blue dots, biological replicates of males at 2 ($n = 5$) and 24 ($n = 5$) months of age. $P = 0.01$ (one-tailed $t$ test). **j** Testis tubule cross-sections in males at 2 (*Left*; a stage XII tubule) and 24 (*Middle* and *Right*; a tubule containing stages X, XI, and XII) months of age, stained with He-PAS. *Upper* (2 and 24 months) and *Lower left* (24 months) panels enlarge black boxed regions in *Lower* (2 months) and *Lower right* (24 months) panels. Green boxed regions (#3 and #6) indicate areas shown in higher magnification in Supplementary Fig. S2a. Scale bars, 20 μm. **k** Number of tubule cross-sections containing more than two stages per total tubule cross-sections. Blue dots, biological replicates of males at 2 ($n = 5$) and 24 ($n = 5$) months of age. $P = 0.002$ (one-tailed $t$ test).

testes, we predicted that sperm numbers stored in cauda epididymides would be correspondingly decreased. We thus counted the sperm numbers in cauda epididymides, which is a widely used and sensitive method of assessing sperm production[31,32]. Indeed, total sperm numbers in cauda epididymides were significantly decreased at 24 months (Fig. 3g, h and Supplementary Fig. 3b), indicating that the increased size of the aged epididymides is due not to sperm numbers but to increased mass of somatic tissues.

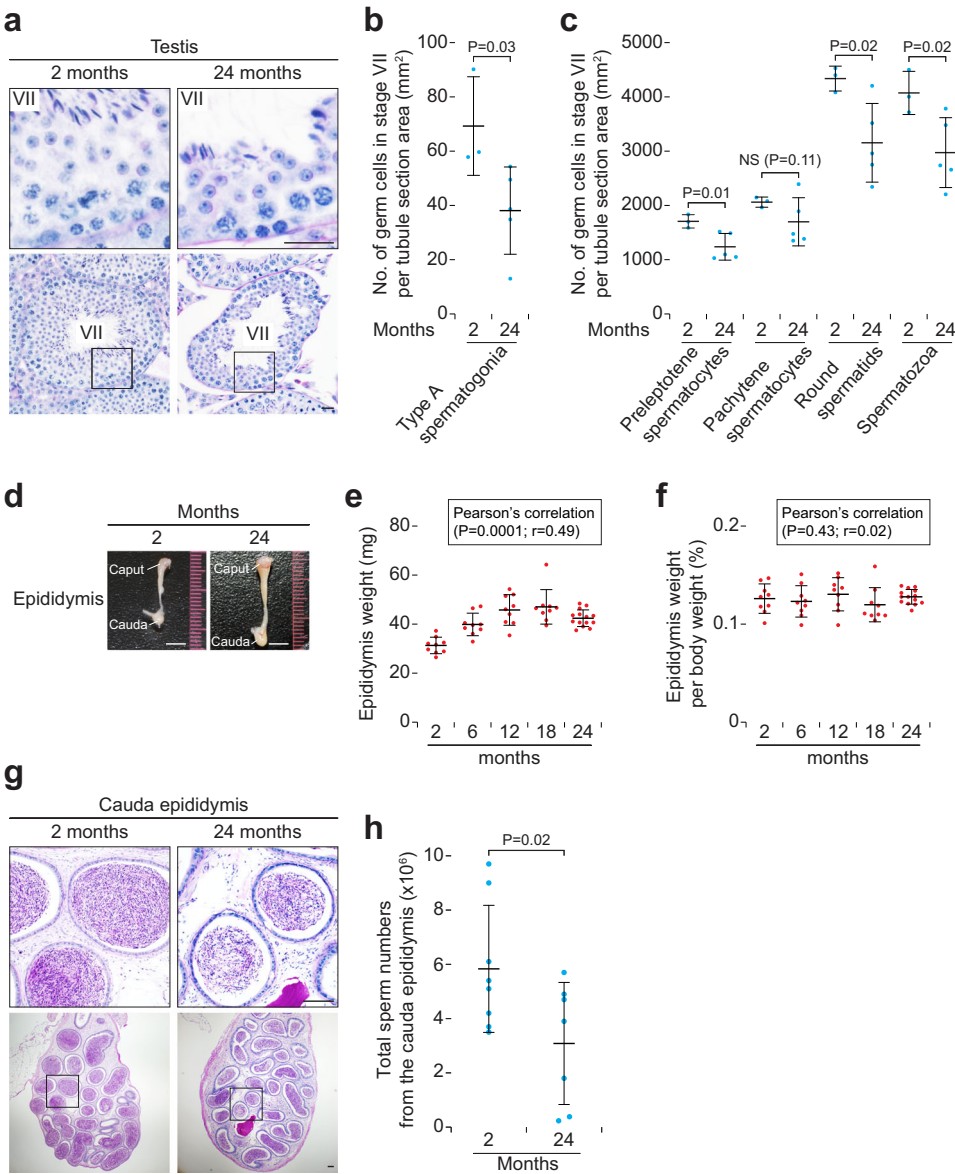

**Fig. 3 Sperm production in testes and sperm numbers stored in epididymides decrease with age. a** Testis tubule cross-sections in stage VII of males at 2 (Left) and 24 (Right) months of age, stained with hematoxylin and periodic acid-Schiff (He-PAS). Upper panels enlarge black boxed regions in Lower panels. Scale bar, 20 μm. **b** Number of germ cells (type A spermatogonia) in stage VII tubule cross-sections per tubule section area ($mm^2$). Blue dots, biological replicates of males at 2 ($n = 3$) and 24 ($n = 5$) months of age. $P = 0.03$ (one-tailed $t$ test). **c** Number of germ cells (preleptotene and pachytene spermatocytes, round spermatids, and spermatozoa) in stage VII tubule cross-sections per tubule section area ($mm^2$). Blue dots, biological replicates of males at 2 ($n = 3$) and 24 ($n = 5$) months of age. $P = 0.02$ and 0.01 (one-tailed $t$ test). NS, not significant ($P > 0.05$; one-tailed $t$ test). **d** Gross morphology of (caput and cauda) epididymides in males at 2 and 24 months of age. Scale bars, 5 mm. **e** Epididymis weight (mg) in males at 2, 6, 12, 18, and 24 months of age. Error bars, mean ± SD. Red dots, epididymis values in males at 2 ($n = 5$), 6 ($n = 5$), 12 ($n = 5$), 18 ($n = 5$), and 24 ($n = 7$) months of age. $P = 0.0001$ and $r = 0.49$ (Pearson's correlation). **f** Epididymis weight per body weights (%) in males at 2, 6, 12, 18, and 24 months of age. Error bars, mean ± SD. Red dots, testis values in males at 2 ($n = 5$), 6 ($n = 5$), 12 ($n = 5$), 18 ($n = 5$), and 24 ($n = 7$) months of age. $P = 0.43$ and $r = 0.02$ (Pearson's correlation). **g** Cauda epididymis tubule longitudinal-sections in males at 2 (Left) and 24 (Right) months of age, stained with He-PAS. *Upper* panels enlarge black boxed regions in Lower panels. Scale bars, 100 μm. **h** Total sperm numbers from the cauda epididymis (×$10^6$). Blue dots, biological replicates of males at 2 ($n = 8$) and 24 ($n = 7$) months of age. $P = 0.02$ (one-tailed $t$ test).

**Decreased proliferation of type A spermatogonia leads to the decreased number of spermatozoa in aged testes.** As mentioned above, the numbers of spermatocytes, spermatids, and spermatozoa were decreased in stage VII at 24 months (Fig. 3c). However, there were no discernible abnormalities in these germ cells at the morphological level (Fig. 3a and Supplementary Fig. 3a), suggesting that the decrease in the numbers of these germ cells was mainly due to the decreased number of type A spermatogonia (Fig. 3b). To verify that type A spermatogonia undergo

differentiation, we tested for key molecular features of spermatogonial differentiation by immunostaining. In the testis, during spermatogonial differentiation in stages VII-VIII, type A spermatogonia express STRA8 and start to enter mitotic S phase[4,27,29]. We confirmed that type A spermatogonia normally expressed STRA8 and incorporated BrdU, a marker of S phase, at 24 months (Fig. 4a and Supplementary Fig. 4a); because STRA8 and 5-bromo-2-deoxyuridine (BrdU) are also functional markers of meiotic initiation[8,9,27], preleptotene spermatocytes were

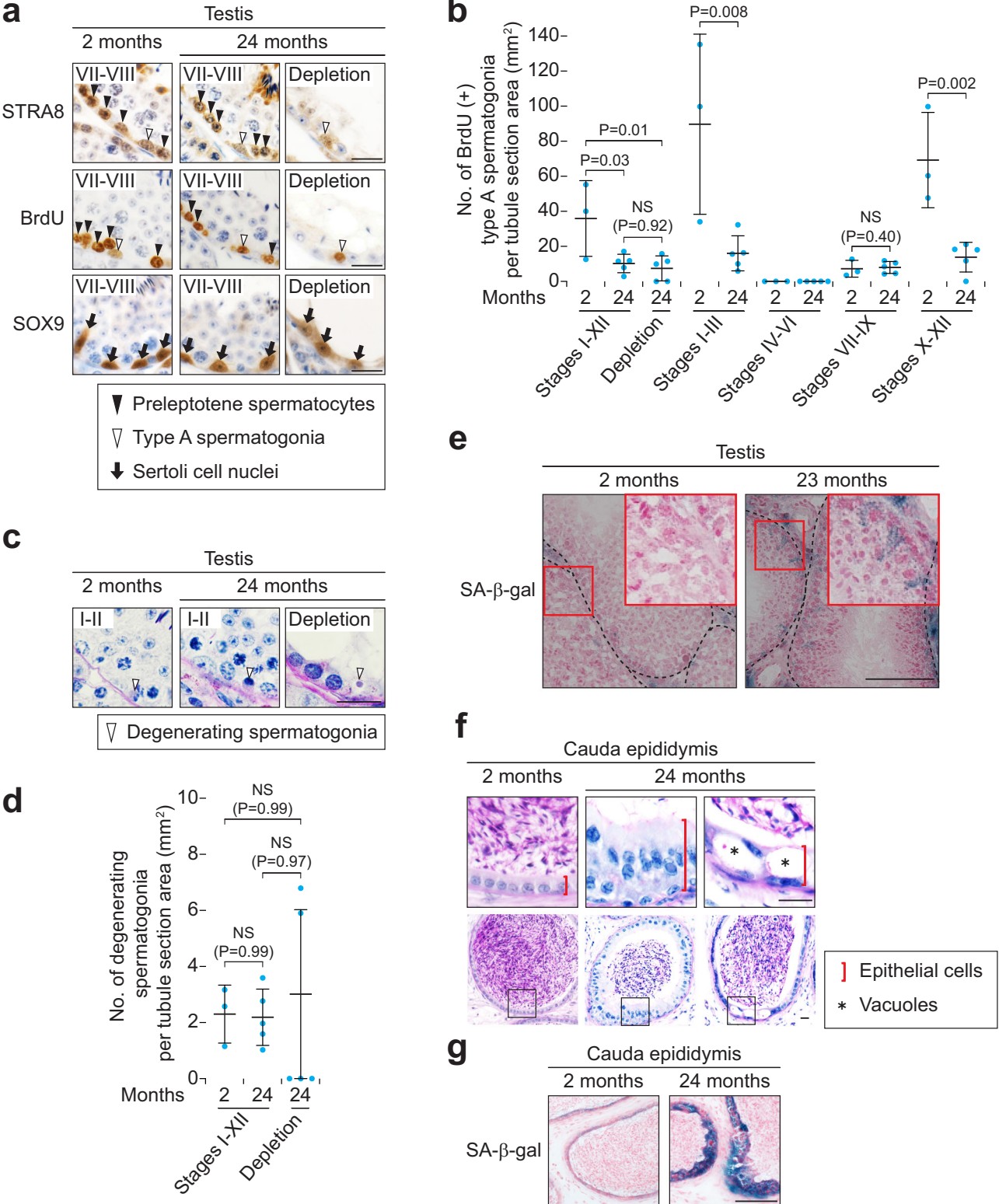

positive for STRA8 and BrdU as expected. Indeed, the number of BrdU-positive type A spermatogonia was not changed in stages VII-IX, when they undergo cell division for their differentiation toward spermatocytes[4,29] (Fig. 4b). By contrast, the BrdU-positive cell numbers were decreased specifically in stages X-XII and I-III, when undifferentiated type A spermatogonia proliferate[4,29] (Fig. 4b). Further, we found that the number of apoptotic degenerating spermatogonia was not changed with age

(Fig. 4c, d). These results indicate that the decreased proliferation of type A spermatogonia simply leads to the decreased number of spermatozoa in aged testes.

**Somatic cells but not germ cells show senescence-associated β-galactosidase activity in aged testes and epididymides.** In most of germ cell-depleted tubules at 24 months, some

**Fig. 4 Somatic cells but not germ cells show senescence-associated β-galactosidase activity in aged testes and epididymides. a** Immunostaining for STRA8, BrdU, and SOX9, with hematoxylin counterstain, on testis cross-sections of males at 2 (Left; a stage VII-VIII tubule) or 24 (Middle, a tubule in stages VII-VIII; Right, a tubule showing germ cell depletion) months of age. Arrows, Sertoli cell nuclei. Arrowheads, type A spermatogonia (white) and preleptotene spermatocytes (black). Scale bars, 20 µm. **b** Number of BrdU-positive type A spermatogonia per tubule section area (mm$^2$). Blue dots, biological replicates of males at 2 ($n = 3$) and 24 ($n = 5$) months of age. $P = 0.03$ and 0.01 (Tukey-Kramer test). NS, not significant ($P > 0.05$; Tukey-Kramer test). $P = 0.008$ and 0.002 (one-tailed $t$ test). NS, not significant ($P > 0.05$; one-tailed $t$ test). **c** Testis tubule cross-sections of males at 2 (Left; a stage I-II tubule) or 24 (Middle, a stage I-II tubule; Right, a tubule showing germ cell depletion) months of age, stained with hematoxylin and periodic acid-Schiff (He-PAS). Arrowheads, degenerating spermatogonia. Scale bar, 20 µm. **d** Number of degenerating spermatogonia per tubule section area (mm$^2$). Blue dots, biological replicates of males at 2 ($n = 3$) and 24 ($n = 5$) months of age. NS, not significant ($P > 0.05$; Tukey-Kramer test). **e** Senescence-associated β-galactosidase (SA-β-gal) staining, with Nuclear Fast Red counterstain, on testis cross-sections of males at 2 (Left) or 23 (Right) months of age. Insets (red) enlarge red boxed regions. Scale bars, 100 µm. **f** Cauda epididymis tubule longitudinal-sections in males at 2 (Left) and 24 (Middle and Right) months of age, stained with hematoxylin and periodic acid-Schiff (He-PAS). Upper panels enlarge black boxed regions in Lower panels. Red bars, epithelial cells. Asterisks, vacuoles. Scale bars, 20 µm. **g** Senescence-associated β-galactosidase (SA-β-gal) staining, with Nuclear Fast Red counterstain, on cauda epididymis tubule longitudinal-sections of males at 2 (Left) or 24 (Right) months of age. Insets (red) enlarge red boxed regions. Scale bars, 100 µm.

---

spermatogonia were present with a few other germ cell types (Fig. 2e and Supplementary Fig. 2b), like other germ cell-depleted testis models[33–36]. The type A spermatogonia in germ cell-depleted tubules were positive for STRA8 and BrdU (Fig. 4a and Supplementary Fig. 4a), suggesting that these type A spermatogonia were competent to express intrinsic factors for differentiation, but failed to undergo spermatogenesis due to extrinsic factors (i.e., the somatic environment). Thus, we immunostained for SOX9, a marker of Sertoli cells. Indeed, the number of Sertoli cells was decreased at 24 months (Fig. 4a and Supplementary Fig. 4a, b). We then tested whether germ cells and/or somatic cells in aged testes show ageing by senescence-associated β-galactosidase (SA-β-gal) staining, a widely used biomarker for cellular senescence both in vivo and in vitro[37–39]. At 23 months, germ cells were negative for SA-β-gal signals, whereas Sertoli cell cytoplasm, which surrounds germ cells, were positive for the signals (Fig. 4e and Supplementary Fig. 4c). Moreover, SA-β-gal signals on the interstitial tissues between tubules were markedly increased at 23 months compared to 2 months. We conclude that somatic cells but not germ cells show SA-β-gal signals in aged testes.

We next tested whether somatic cells in aged epididymides show histological abnormalities and SA-β-gal activities. In 24-month-old cauda epididymides, epithelial cell layers of tubules were thickened and contained some vacuoles (Fig. 4f); these epithelial cells had strong SA-β-gal signals compared to 2-month-old epithelial cells, whereas spermatozoa in tubules were negative for the signals (Fig. 4g and Supplementary Fig. 4d). We conclude that somatic cells in the aged epididymis show histological abnormalities and strong SA-β-gal signals.

**RNA-Seq analysis in aged testes and epididymides shows up-regulation of genes related to inflammation and senescence-associated secretory phenotype factors.** To analyze mRNA expressions in aged testes and epididymides at genome-wide level, we performed mRNA-Seq on these samples at 2 and 24 months (Fig. 5a); in 24-month-old testes, 422 genes were up-regulated (fold change ≥2), and 790 genes were down-regulated (fold change ≤ −2); in 24-month-old cauda epididymides, 1086 genes were up-regulated, and 942 genes were down-regulated. Using these up-regulated and down-regulated gene lists, we performed Gene Ontology (GO) analysis. In 24-month-old testes and cauda epididymides, most of the top 10 enriched GO terms (biological process) of up-regulated genes were related to inflammation (Fig. 5b and Supplementary Fig. 5a), which is the major ageing phenotype in body tissues[40–42]. Indeed, heatmap of differentially expressed genes showed that the genes related to cellular senescence, which includes senescence-associated secretory phenotype (SASP) factors known to stimulate inflammation[43], tended to be up-regulated in aged testes and epididymides (Fig. 5c). We conclude

that mRNA expressions in both aged testes and epididymides show ageing features at genome-wide level.

We next tested for genome-wide expressions of germ cell-related genes using the RNA-Seq data. In 24-month-old testes and cauda epididymides, down-regulated genes related to spermatogenesis or germ cells were absent in the lists of the top 10 enriched GO terms (Supplementary Fig. 5b). Given that germ cells in the testis undergo transcriptional repression during late spermatogenesis (called spermiogenesis), and spermatozoa are transcriptionally inactive in the epididymis[44,45], we focused on aged testes to perform heatmap analysis of key marker genes for spermatogenesis or germ cells. We confirmed that most of the genes for undifferentiated spermatogonia, spermatogonial differentiation, meiosis, and spermiogenesis were not down-regulated at 24 months of age (Fig. 5c). These RNA-Seq data were consistent with histological and immunohistochemical data showing no discernible abnormalities in germ cells (Figs. 3a and 4a and Supplementary Figs. 3a and 4a). Moreover, key marker genes for Sertoli cells and Leydig cells were also not down-regulated (Fig. 5c), suggesting that aged Sertoli cells and Leydig cells do not impair these expressions at the mRNA level.

**Aged epididymides accumulate structurally abnormal spermatozoa with decreased motility.** Having found that aged epididymides show histological abnormalities and ageing features, we hypothesized that the spermatozoa stored in aged epididymides should undergo morphological and/or functional disorders. As predicted, the spermatozoa at 24 months had abnormal morphologies including bent head or bent (or broken) midpiece (Fig. 6a); the abnormal spermatozoa were more accumulated in cauda (34.9%) than in caput (9.1%) epididymides at 24 months ($P < 0.00001$; $\chi^2$ test; Fig. 6b). We then performed ultrastructural analysis of cauda epididymal spermatozoa by SEM. The spermatozoa at 24 months exhibited structurally abnormal flagella (Fig. 6c and Supplementary Fig. 6a); the midpiece was abnormally curved next to the sperm head or broken at the annulus with a split mitochondrial alignment. Moreover, some spermatozoa at 24 months had midpieces that lack mitochondria (Supplementary Fig. 6b).

To test whether motility was affected in aged spermatozoa with the structurally abnormal flagella, we analyzed flagellar bending patterns of spermatozoa. Abnormal spermatozoa at 24 months exhibited inflexible movement after incubation in capacitation media for both 10 min and 120 min, when spermatozoa have undergone hyperactivation[46] (Fig. 6d and Supplementary Fig. 6c). Moreover, motility, progressive motility, and velocity parameters of spermatozoa at 24 months were lower than those at 2 months (Fig. 6e, f and Supplementary Fig. 6d). We conclude that aged epididymides accumulate structurally abnormal spermatozoa with decreased motility.

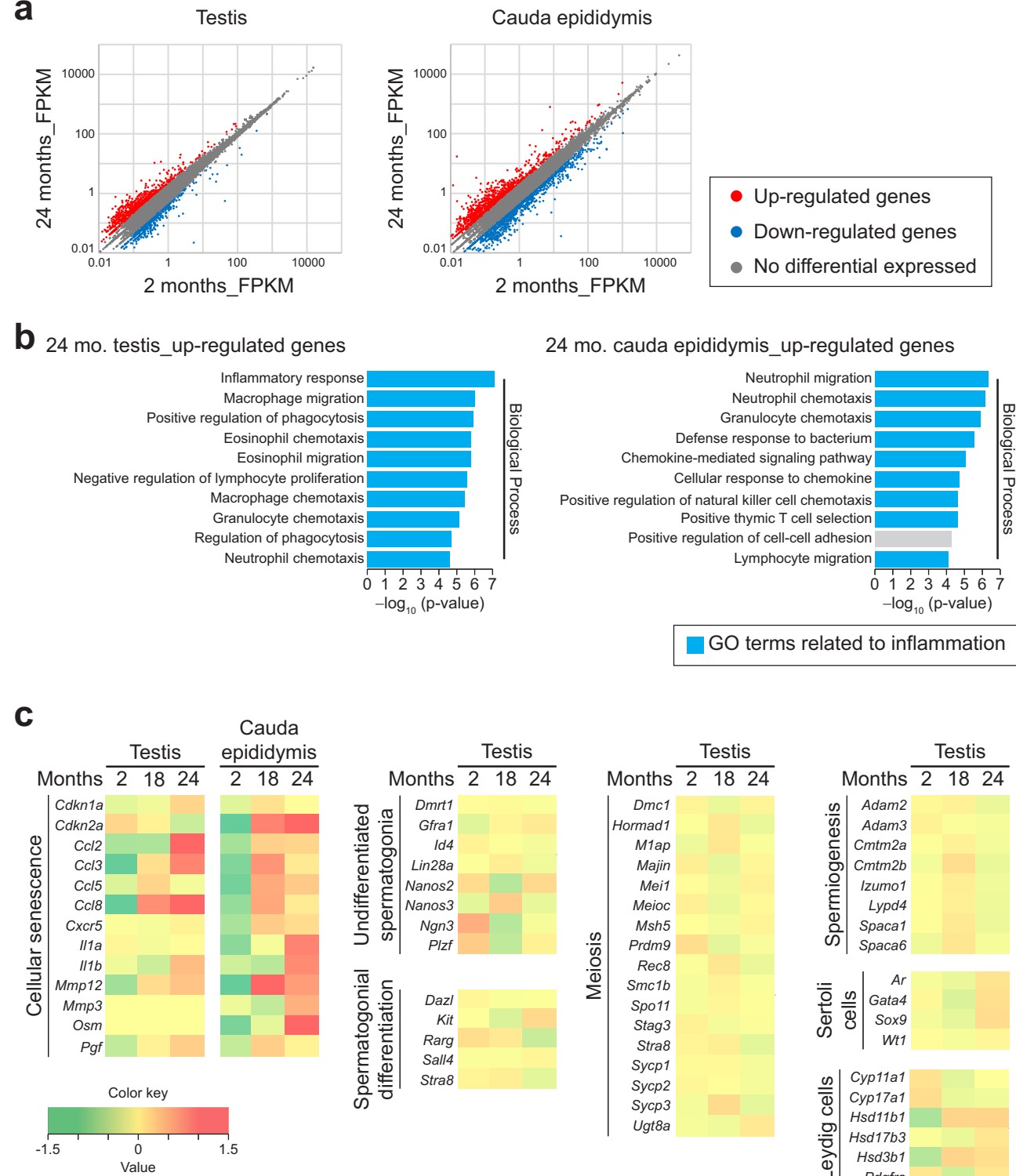

**Fig. 5 RNA-Seq analysis in aged testes and epididymides shows up-regulation of genes related to inflammation and senescence-associated secretory phenotype factors. a** Scatter plots of RNA-Seq data comparing the normalized FPKM values for all genes in testes (*Left*) or cauda epididymides (*Right*) of males at 2 (x-axis, *n* = 5) vs. 24 (y-axis, *n* = 5) months of age. Red dots, up-regulated genes (fold change ≥2). Blue dots, down-regulated genes (fold change ≤−2). Gray dots, no differential gene expression (−2 <fold change <2). **b** Gene Ontology (GO) terms (biological process) of up-regulated genes (fold change ≥2) on RNA-Seq data in testes (*Left*) or cauda epididymides (*Right*) of males at 2 (*n* = 5) vs. 24 (*n* = 5) months of age. The top 10 enriched terms are displayed based on the -log10 (P-value). Y-axis, −log10 (P-value). Blue graphs, GO terms related to inflammation. **c** Heatmap of differentially expressed genes related to cellular senescence, undifferentiated spermatogonia, spermatogonial differentiation, meiosis, spermiogenesis, Sertoli cells, and Leydig cells, on RNA-Seq data in testes or cauda epididymides of males at 2 (*n* = 5) vs. 18 (*n* = 5) vs. 24 (*n* = 5) months of age.

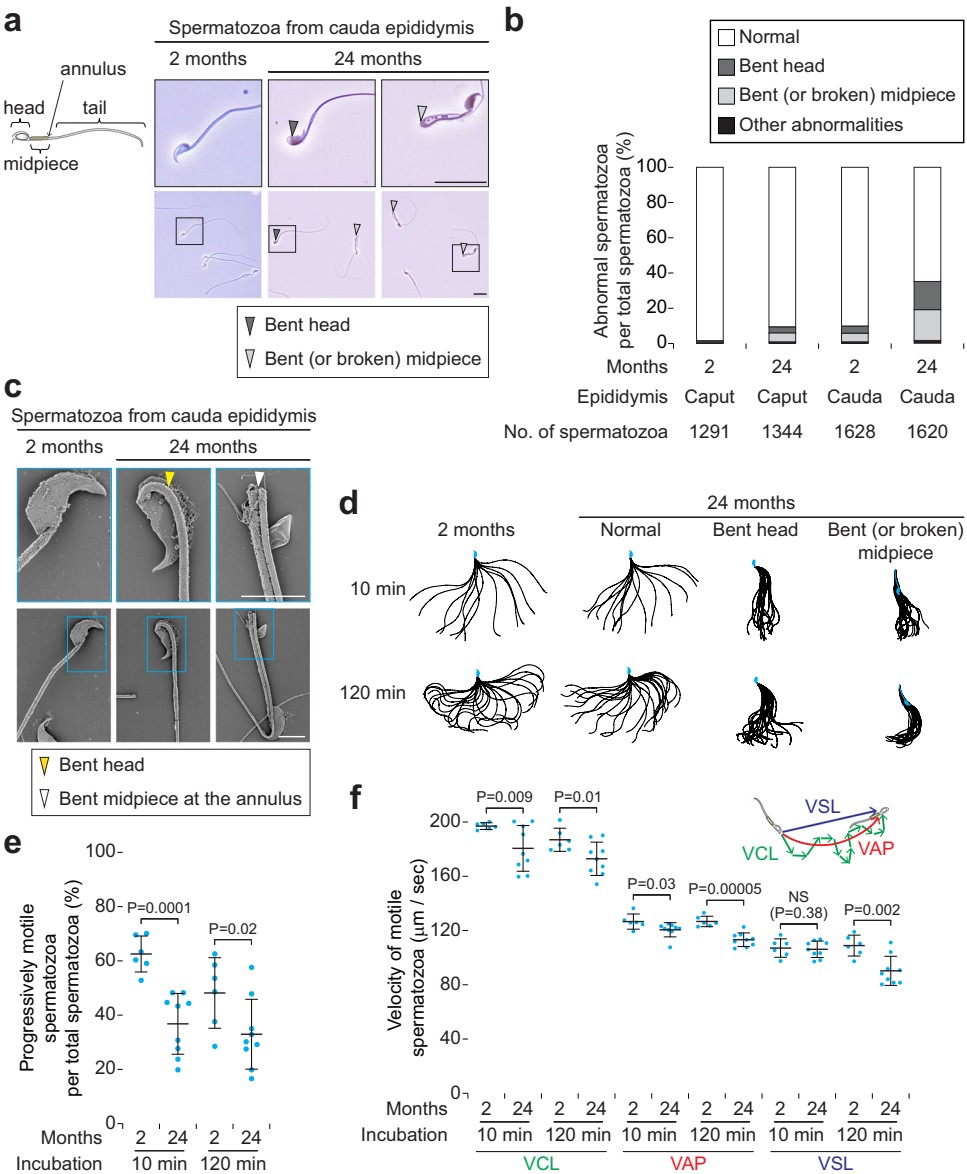

**Fig. 6 Aged epididymides accumulate abnormal spermatozoa with decreased motility. a** Morphology of cauda epididymal spermatozoa in males at 2 (Left) and 24 (Middle and Right) months of age. Upper panels enlarge black boxed regions in Lower panels. Arrowheads, bent head (dark gray) and bent (or broken) midpiece (light gray). Scale bars, 20 μm. **b** Number of abnormal spermatozoa per total spermatozoa (%), collected from caput and cauda epididymides, in males at 2 ($n = 5$ biological replicates) and 24 ($n = 6$ biological replicates) months of age. Percentages of abnormal spermatozoa were increased from 1.2% (16/1291 at 2 months) to 9.1% (122/1344 at 24 months) in caput epididymides ($P < 0.00001$; $\chi^2$ test), and from 9.1% (158/1628 at 2 months) to 34.9% (566/1620 at 24 months) in cauda epididymides. Boxes, normal (white), bent head (dark gray), bent or broken midpiece (light gray), and other abnormalities (black). **c** Ultrastructural analysis of cauda epididymal spermatozoa, in males at 2 (Left) and 24 (Middle and Right) months of age, by scanning electron microscopy (SEM). Upper panels enlarge blue boxed regions in Lower panels. Arrowheads, bent head (yellow) and bent midpiece at the annulus (white). Scale bars, 5 μm. **d** Flagellar bending patterns of spermatozoa at 10 min (Upper) and 120 min (Lower) after incubation in capacitation media, in males at 2 (Far left) and 24 (Center left, normal; Center right, bent head; Far right, bent or broken midpiece) months of age. Single frames throughout one beating cycle or 20 frames are superimposed. **e** Progressively motile spermatozoa per total spermatozoa (%), at 10 min and 120 min after incubation in capacitation media, in males at 2 and 24 months of age. Error bars, mean ± SD. Blue dots, biological replicates of males at 2 ($n = 6$) and 24 ($n = 9$) months of age. $P = 0.02$ and 0.0001 (one-tailed $t$ test). **f** Velocity parameters of motile spermatozoa (μm/sec) at 10 min and 120 min after incubation in capacitation media, in males at 2 and 24 months of age. VCL curvilinear velocity, VAP average path velocity, VSL straight line velocity. Error bars, mean ± SD. Blue dots, biological replicates of males at 2 ($n = 6$) and 24 ($n = 9$) months of age. $P = 0.03$, 0.02, 0.01, 0.009, and 0.00005 (one-tailed $t$ test). NS, not significant ($P > 0.05$; one-tailed $t$ test).

**Spermatozoa in aged epididymides adversely affect in vitro fertilization and subsequent early embryonic development**. To evaluate the fertilizing ability of aged spermatozoa in the epididymis, we performed in vitro fertilization (IVF). The number of fertilized eggs per total oocytes was significantly decreased when 24-month-old spermatozoa were used for IVF (Fig. 7a). The number of two pronuclear (2PN) eggs per fertilized eggs tended to increase with age (Fig. 7b), indicating that aged spermatozoa do not cause polyspermic (≥3PN) fertilization.

After IVF, we cultured fertilized eggs (embryos) and monitored their development from 1-cell (Day 0) to blastocyst stages (Day 4). Morula and blastocyst formation rates were significantly

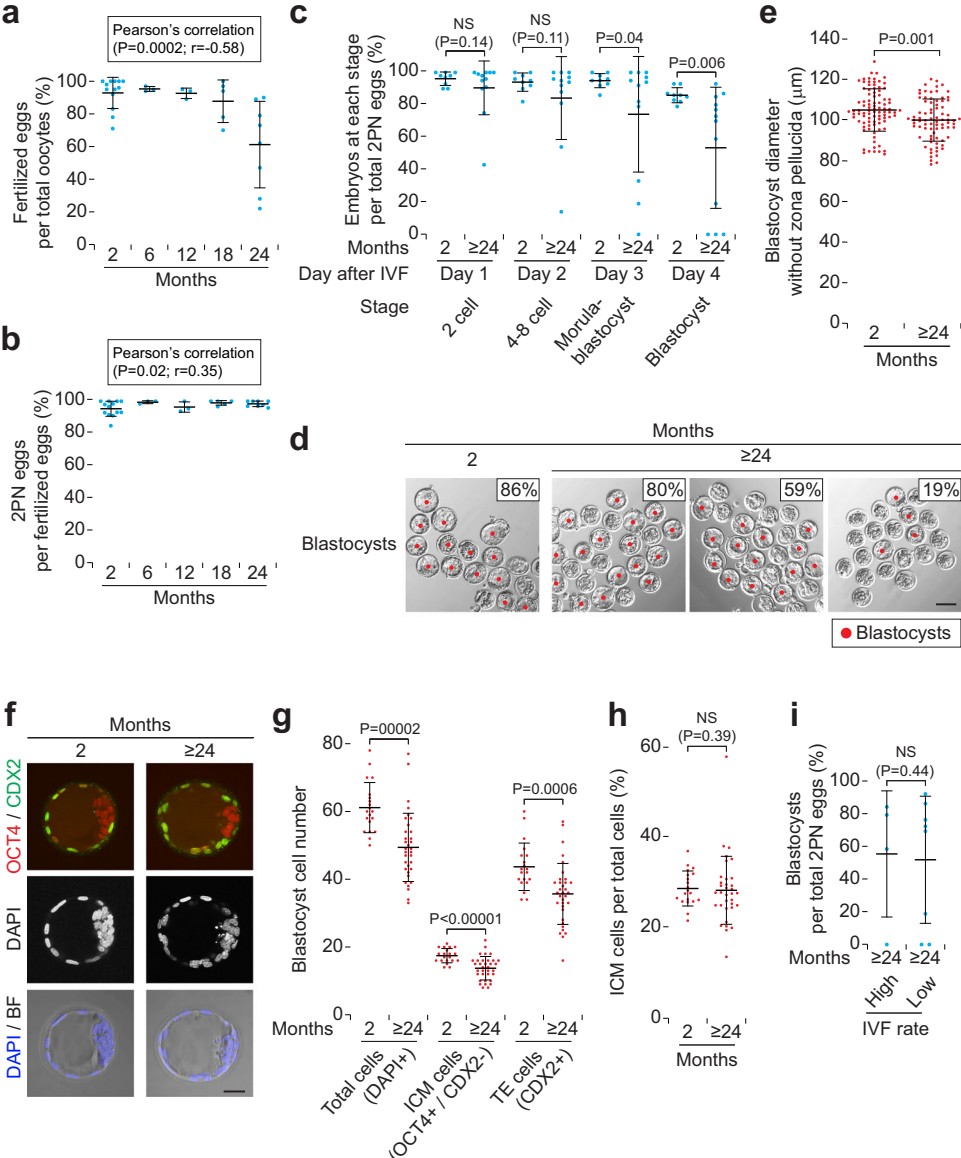

**Fig. 7 Spermatozoa in aged epididymides adversely affect in vitro fertilization and subsequent early embryonic development. a** Number of fertilized eggs per total oocytes (%) at 8 h after in vitro fertilization (IVF), by using spermatozoa in males at 2, 6, 12, 18, and 24 months of age. Error bars, mean ± SD. Blue dots, biological replicates of males at 2 ($n = 13$), 6 ($n = 3$), 12 ($n = 3$), 18 ($n = 5$), and 24 ($n = 8$) months of age. $P = 0.0002$ and $r = -0.58$ (Pearson's correlation). **b** Number of two pronuclear (2PN) eggs per fertilized eggs (%), at 8 h after IVF, by using spermatozoa in males at 2, 6, 12, 18, and 24 months of age. Error bars, mean ± SD. Blue dots, biological replicates of males at 2 ($n = 13$), 6 ($n = 3$), 12 ($n = 3$), 18 ($n = 5$), and 24 ($n = 8$) months of age. $P = 0.02$ and $r = 0.35$ (Pearson's correlation). **c** Number of embryos at each stage per total 2PN eggs (%), at Day 1–4 after IVF, produced using spermatozoa from males at 2 and ≥24 months of age. Error bars, mean ± SD. Blue dots, biological replicates of males at 2 ($n = 9$) and ≥24 ($n = 12$) months of age. $P = 0.04$ and 0.006 (one-tailed $t$ test). NS, not significant ($P > 0.05$; one-tailed $t$ test). **d** Blastocysts, at Day 4 after IVF, produced using spermatozoa from males at 2 (Far left) and ≥24 (Center left, Center right, and Far right) months of age. Insets, number of embryos at blastocyst stage per total 2PN eggs (%). Red dots, blastocysts. Scale bars, 100 μm. **e** Blastocyst diameter without zona pellucida (μm), at Day 4 after IVF, using spermatozoa from males at 2 ($n = 5$) and ≥24 ($n = 5$) months of age. Error bars, mean ± SD. Each red dot represents a blastocyst ($n = 91$, at 2 months; $n = 79$, at ≥24 months). $P = 0.001$ (one-tailed $t$ test). **f** Blastocysts at Day 4 after IVF, using spermatozoa from males at 2 (Left) and ≥24 (Right) months of age, immunostained for OCT4 and CDX2 (merged), with DAPI counterstain. BF, bright field, were merged with DAPI. Z-stack images (10–12 μm thickness) were converted into single projection images. Scale bars, 30 μm. **g** Blastocyst cell number, at Day 4 after IVF, using spermatozoa from males at 2 ($n = 3$) and ≥ 24 ($n = 5$) months of age. Each red dot represents a blastocyst ($n = 20$, at 2 months; $n = 35$, at ≥24 months). $P = 0.0006$ and 0.0002, and <0.00001 (one-tailed $t$ test). Total cells, DAPI-positive cells. Inner cell mass (ICM) cells, OCT4-positive and CDX2-negative cells. Trophectoderm (TE) cells, CDX2-positive cells. **h** Number of ICM cells per total cells (%) in blastocysts at Day 4 after IVF, by using spermatozoa in males at 2 ($n = 3$) and ≥24 ($n = 5$) months of age. Each red dot represents a blastocyst ($n = 20$, at 2 months; $n = 35$, at ≥24 months). NS, not significant ($P > 0.05$; one-tailed $t$ test). **i** Number of blastocysts per total 2PN eggs (%), at Day 4 after IVF (**c**), using spermatozoa from males at ≥24 months of age. Error bars, mean ± SD. Blue dots, biological replicates of males at ≥24 ($n = 12$) months of age, were divided into two groups: high (90–73%) and low (71–22%) IVF rates at Day 0. NS, not significant ($P > 0.05$; one-tailed $t$ test).

decreased in embryos derived from spermatozoa at ≥24 (24–26) months (Fig. 7c, d); at Day 4, the decrease in the number of blastocysts was due not to delayed development, but to degeneration or fragmentation of embryos (Fig. 7d and Supplementary Fig. 7). Moreover, the diameters of blastocysts derived from ≥24-month-old spermatozoa were significantly decreased (Fig. 7e). We confirmed that these blastocyst cell numbers, including inner cell mass (ICM) and trophectoderm (TE) cells, were decreased, using markers OCT4 and CDX2[47,48] (Fig. 7f, g); the number of ICM cells per total cells was comparable between 2 and ≥24-month-old spermatozoa (Fig. 7h).

We tested whether low IVF rates by aged spermatozoa are correlated with low embryonic developmental rates. There were no differences in blastocyst formation rates between the embryos derived from high (90-73%) and low (71–22%) IVF rates by spermatozoa at ≥24 months (Fig. 7i), indicating that low IVF rates and low embryonic developmental rates are caused independently by aged spermatozoa.

**Both somatic cells and spermatozoa in aged epididymides accumulate DNA damage**. Because low blastocyst formation rates are correlated with sperm DNA damage that does not affect sperm morphology or fertilization[49–52], we analyzed DNA damage in aged testes and epididymides. We first immunostained for γH2AX (phosphorylated H2A histone family member X), which is a sensitive molecular marker of DNA damage (double-strand break)[53]. In 2-month-old testes, strong γH2AX signals were detected in the preleptotene spermatocytes undergoing meiotic DNA double-strand breaks and in sex bodies of pachytene spermatocytes (Fig. 8a and Supplementary Fig. 8a), as previously reported[27,54]. Localization patterns of γH2AX signals were comparable between 2 and 24-month-old testes; no ectopic signals were detected in germ cells or Sertoli cells at 24 months (Fig. 8a and Supplementary Fig. 8a). By contrast, γH2AX signals were detected in tubular epithelial and interstitial cells of 24-month-old cauda epididymides, whereas there were no signals in epididymides at 2 months (Fig. 8b and Supplementary Fig. 8b).

We next tested whether speramtozoa stored in aged epididymides undergo DNA damage. In spermatozoa, most histones including H2AX, are replaced with protamines[55]. We thus measured the levels of DNA damage using the comet assay, which is a single-cell gel electrophoresis technique to detect DNA fragments in spermatozoa[56,57]. Sperm comet tail lengths, which represent fragmented DNAs, were significantly increased at ≥ 24 months (Fig. 8c, d). We conclude that spermatozoa in aged epididymides accumulate DNA damage.

## Discussion

We investigated the effect of ageing on male fertility and germ cells in testes and epididymides. We now show that in vivo male fertility decreases with age, that overall sperm production in aged testes is decreased, and that the spermatozoa in aged epididymides exhibit abnormal morphology and structure, decreased motility, increased DNA damage, and low fertilization and developmental rates. Here we propose that these multiple ageing effects on germ cells lead to decreased in vivo male fertility (Fig. 8e).

We find that the frequency of mating behavior and (both serum and testicular) testosterone levels are not decreased until 24 months of age. By contrast, previous reports showed the decreased serum testosterone levels with age in mice[20,58]. This difference between previous reports and our finding likely reflects differences in strains/genetic background of mice and/or housing environments; in our C57BL/6 J mouse model, their average lifespan is 29 months[59]. Testosterone is required for the

development of spermatocytes and spermatids during meiotic progression and spermiogenesis, respectively[60,61]. Indeed, there were no morphological abnormalities or developmental arrests in spermatocytes or spermatids at 24 months in our mouse model, indicating that testosterone levels in aged mice are high enough to maintain spermatogenesis. Nevertheless, in vivo fertility decreases in aged males. Our findings provide evidence that aged males undergo testosterone-independent functional decline in testes and epididymides, resulting in the decreased in vivo fertility.

In aged males, we find that the efficiency of sperm production is impaired in whole testes. Specifically, about 20% of seminiferous tubules showed abnormal stages (germ cell associations): germ cell depletion, sperm release failure, and perturbed stages (more than two stages in a tubule cross-section area). The remaining 80% of aged tubules with normal stages contained lower number of germ cells compared to young tubules. The decreased number of undifferentiated spermatogonia, caused by their decreased proliferation, is consistent with previous SSC transplantation studies showing decreased SSC numbers and activity in aged testes[15,16]. We now find that during and after spermatogonial differentiation, germ cells in aged testes show no morphological abnormalities or developmental arrests from differentiating spermatogonia to spermatozoa. Indeed, key marker genes for spermatogenesis are not down-regulated in aged testes at genome-wide level. We thus conclude that decreased number of spermatozoa in aged seminiferous tubules with normal stages is simply due to the decreased proliferation of undifferentiated spermatogonia.

Transplantation of young SSCs into aged testes has revealed that the function of the somatic environment declines with age[15]. We now provide several observations that support the declined function of somatic environment in aged testes. First, somatic cells but not germ cells show an ageing feature, SA-β-gal signal, in aged testes. Second, our observation that aged testes partially exhibit germ cell depletion, sperm release failure, and perturbed stages, accord with published phenotypes of Sertoli Cell dysfunctions by genetic ablations[11,62,63]. Further, type A spermatogonia in germ cell-depleted aged tubules expressed STRA8 and entered S phase, which are key functional features of spermatogonial differentiation, suggesting that these spermatogonia are intrinsically competent to undergo differentiation and subsequent spermatogenesis. Finally, we show that the number of Sertoli cells is decreased in aged testes. The decreased Sertoli cell numbers may be involved in decreased number of germ cells and/or abnormal seminiferous stages of aged testes because Sertoli cell numbers determine the capacity of germ cells and sperm production in the tubule[64–66]. Further studies are needed to determine how aged Sertoli cells functionally affect germ cells and spermatogenesis.

We do not yet know whether (and how) other somatic cell types in aged testes and epididymides affect sperm production and sperm function, respectively. We find that strong SA-β-gal signals on the interstitial tissues in aged testes and tubular epithelial cells in aged epididymides. Moreover, in both aged testes and epididymides, genes related to inflammation and cellular senescence are up-regulated at genome wide level. The simplest interpretation of these observations is that interstitial tissues in aged testes and tubular epithelial cells in aged epididymides undergo inflammation and cellular senescence. Indeed, the interstitial tissues in testes comprise blood vessels, T cells, and dendritic cells, all of which are known to undergo inflammation and senescence with age[67–70], and the epithelial cells in aged epididymides show morphological abnormalities and DNA damage. Although interstitial tissues in testes also comprise Leydig cells, which produce testosterone, functional decline of testosterone production has yet to occur at least at 24 months of

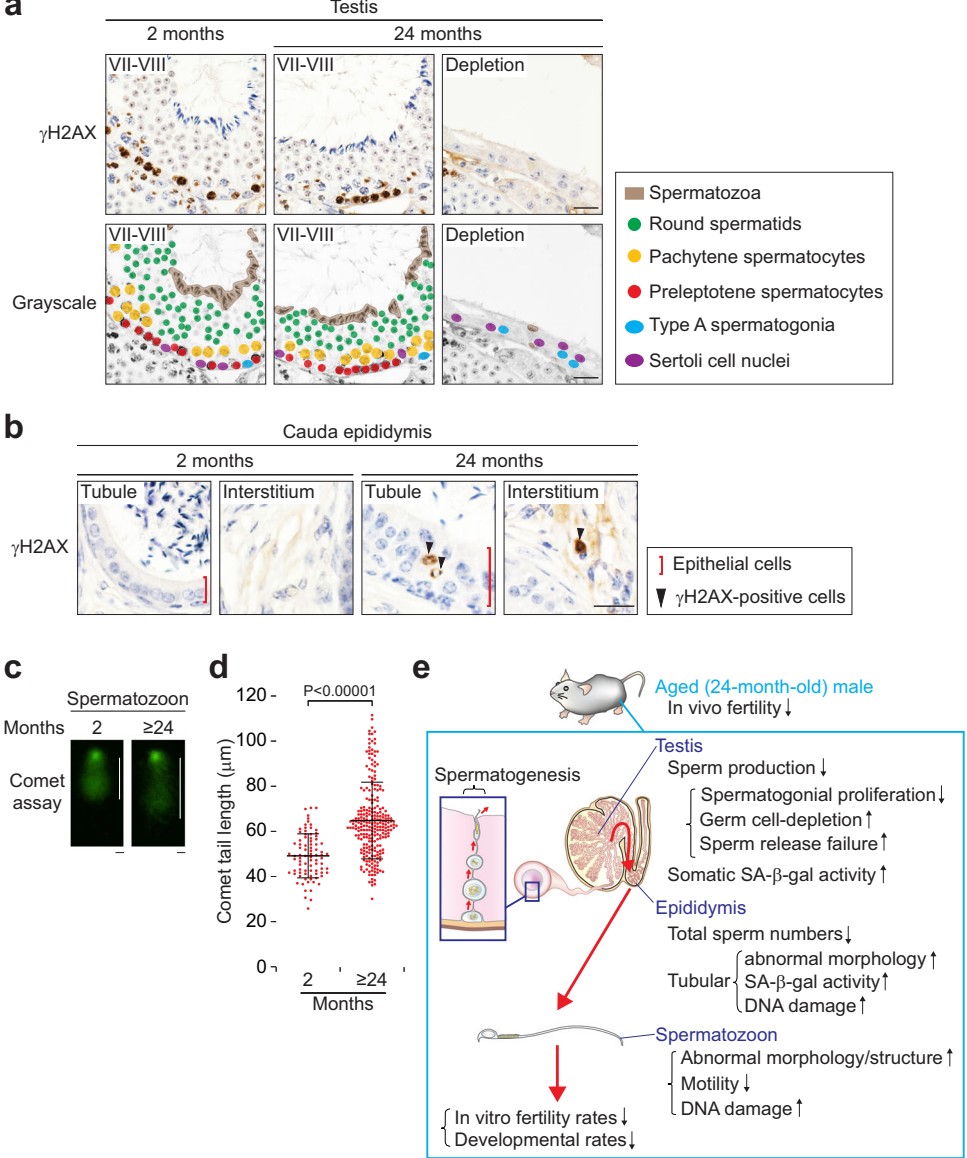

**Fig. 8 Both somatic cells and spermatozoa in aged epididymides accumulate DNA damage. a** Immunostaining for γH2AX, with hematoxylin counterstain, on testis cross-sections of males at 2 (Left, a stage VII-VIII tubule) or 24 (Middle, a tubule in stages VII-VIII; Right, a tubule showing germ cell depletion) months of age. Lower panels are the grayscale versions of Upper panels. Dots, Sertoli cell nuclei (purple), type A spermatogonia (blue), preleptotene spermatocytes (red), pachytene spermatocytes (yellow), and round spermatids (green). Brown areas, spermatozoa. Scale bars, 20 μm. **b** Immunostaining for γH2AX, with hematoxylin counterstain, on cauda epididymis tubule longitudinal-sections of males at 2 (Far left, a tubule area; Center left, an interstitial space) or 24 (Center right, a tubule area; Far right, an interstitial space) months of age. Red bars, epithelial cells. Arrowheads, γH2AX-positive cells. Scale bar, 20 μm. **c** Comet assay images using spermatozoa from males at 2 ($n = 3$) and ≥24 ($n = 6$) months of age. White bars, comet tail lengths, represent the fragmented sperm DNAs. Scale bars, 10 μm. **d** Comet tail length (μm), using spermatozoa from males at 2 ($n = 3$) and ≥24 ($n = 6$) months of age. Error bars, mean ± SD. Each red dot represents a spermatozoon ($n = 85$, at 2 months; $n = 250$, at ≥24 months). $P < 0.00001$ (one-tailed $t$ test). **e** A proposed model for ageing effects on in vivo male fertility and on germ cells and somatic cells in testes and epididymides.

age. It will be of great interest to determine the somatic cell types that undergo functional decline and impair sperm production and function (directly or indirectly) with age.

We find that aged epididymides contain morphologically and structurally abnormal spermatozoa; the abnormal spermatozoa were more abundant in cauda epididymides than in caput epididymides. The abnormalities in aged spermatozoa restricted their movement, thus resulting in decreased sperm motility. We postulate that both morphological/structural abnormalities and decreased motility in aged spermatozoa cause decrease fertilization rates, based on published data of abnormal spermatozoa[71–73]. We also find that low fertilization rates in aged

spermatozoa are not correlated with low developmental rates, indicating that aged spermatozoa have an impairment other than abnormal morphology/structure and decreased motility. Indeed, we provide evidence that aged spermatozoa accumulate DNA damage, which is known to decrease developmental and pregnancy rates after fertilization[49–52]. In contrast to aged epididymides, we did not detect obvious DNA damage signals on germ cells or somatic cells in aged testes. This finding suggests that aged epididymides may accumulate much more DNA damage than aged testes.

In summary, our findings demonstrate that decreased sperm production in aged testes and functional defects of spermatozoa

in aged epididymides lead to decreased in vivo male fertility. In humans, male fertility is thought to decline with age; semen parameters including semen volume, sperm motility, and normal sperm morphology, start to decline after 35 years of age, and these declines are remarkable over 50 years of age[3]. Specifically, when comparing the semen quality of a 50-year-old man to a 30-year-old man, there was a 3–22% decrease in semen volume, a 4–22% decrease in normal sperm morphology, and a 3–37% decrease in sperm motility[74]. However, the current knowledge on aging of men's fertility is limited because it is largely obtained from non-invasive semen analysis. Moreover, the decreased fertility in older men can be caused not only by ageing but also by diseases or exposure to environmental factors such as chemicals and nutrition. Our histological- and molecular-based findings derived from a C57BL/6 J mouse model are useful to better understand the basic mechanism behind the ageing effect on male fertility in humans as well as other mammals.

## Methods

**Animals**. All animal experiments were approved by the Animal Care and Use Committee of the Research Institute for Microbial Diseases, Osaka University. We have complied with all relevant ethical regulations for animal use. Wild-type C57BL/6 J mice were purchased from Charles River Laboratories Japan, Inc. (Yokohama, Japan) and CLEA Japan, Inc. (Tokyo, Japan). Unless otherwise noted, experiments were performed on male mice at 2, 6, 12, 18, 24, and ≥24 (from 24 to 26) months and female mice at 2 months of age; 24 or 24–26 months of age was used as one of the aged time points, based on previous reports of ageing research[75–77] and on the fact that average lifespan of C57BL/6 J mice is 29 months[59].

**5-bromo-2-deoxyuridine incorporation**. Males received i.p. injections of 10 μL/g body weight of 10 mg/mL BrdU (Sigma-Aldrich, USA) in PBS, 4 h before they were sacrificed.

**Mating test**. Single 2-month-old males were caged with two of 2-month-old females for 4 months, and the females were replaced with new 2-month-old females every 4 months, until the males reached the age of 24 months. The numbers of pups, litters, and vaginal plugs in females were counted in each cage.

**Serum testosterone measurement**. Blood samples were placed in 1.5 mL tubes at room temperature for 30 min, and centrifuged at $1000 \times g$ for 30 min at 4 °C to collect serum. The serum testosterone concentrations were measured by a Parameter Testosterone Assay Kit (KGE010, R&D Systems, USA) according to the manufacturer's protocol.

**Testicular testosterone measurement**. Testicular testosterone concentrations were measured as reported previously[78,79]. In brief, a mouse testis was homogenized in 1 mL of distilled water. As internal standards, Testosterone-$^{13}C_3$ was added to mouse testis suspensions. Testosterone was extracted with 4 mL of methyl tert-butyl ether. After the organic layer was evaporated, the extract was dissolved in 0.5 mL of methanol and diluted with 1 mL of distilled water. The sample was applied to OASIS MAX cartridge which had been successively conditioned with 3 mL of methanol and 3 mL of distilled water. After the cartridge was washed with 1 mL of distilled water, 1 mL of methanol/distilled water/acetic acid (45:55:1, v/v/v), and 1 mL of 1% pyridine solution, the testosterone was eluted with 1 mL of methanol/pyridine (100:1, v/v). After evaporation, the residue was reacted with 50 μL of mixed solution (80 mg of 2-methyl-6-nitrobenzoic anhydride, 20 mg of 4-dimethylaminopyridine, 40 mg of picolinic

acid, and 10 μL of triethylamine in 1 mL of acetonitrile) for 30 min at room temperature. After the reaction, the sample was dissolved in 0.5 mL of ethyl acetate/hexane/acetic acid (15:35:1, v/v) and the mixture was applied to InertSep SI cartridge which had been successively conditioned with 3 mL of acetone and 3 mL of hexane. The cartridge was washed with 1 mL of hexane, and 2 mL of ethyl acetate/hexane (3:7, v/v). Testosterone was eluted with 2.5 mL of acetone/hexane (7:3, v/v). After evaporation, the residue was dissolved in 0.1 mL of acetonitrile/distilled water (2:3, v/v) and the solution was subjected to a LC-MS/MS. The limit of quantification of testosterone was 1 pg per tube. All measurements were performed at ASKA Pharma Medical (Fujisawa, Japan).

**Histology**. Testes and epididymides were fixed overnight in Bouin's solution, embedded in paraffin, sectioned, and stained with hematoxylin and periodic acid-Schiff (PAS). All sections were observed under a light microscope. Germ cell types were identified by their location, nuclear size, and chromatin pattern[80]. Degenerating spermatogonia were identified by their location, nuclear size and morphology, and cell shrinkage[36,80]. Seminiferous tubule stages were determined using hematoxylin and PAS-stained cross sections, according to morphological criteria[25,80]. In brief, the 12 stages were identified primarily based on the first 12 steps of spermatid development. When the patterns of germ cell associations were changed in males at 24 months of age, the stages were identified according to spermatid development[25]. Seminiferous tubule cross-section areas were measured by ImageJ software.

**Sperm count and morphology**. The cauda epididymal spermatozoa were dispersed in PBS, and then their numbers were counted, or their morphology was observed, under a phase-contrast microscope (BX50, Olympus, Tokyo, Japan).

**Ultrastructural analysis of spermatozoa**. Ultrastructural analysis of cauda epididymal spermatozoa by scanning electron microscopy (SEM) was performed as described previously[81]. In brief, cauda epididymal spermatozoa were incubated in TYH medium to disperse, collected in a tube, washed with 0.1 M phosphate buffer (pH 7.4), mounted on coverslips, and fixed with 1% glutaraldehyde in 0.1 M phosphate buffer on ice. After washing, the specimens were postfixed with 1% osmium tetroxide in 0.1 M phosphate buffer containing 1% potassium ferrocyanide, conductive-stained with 1% tannic acid solution and 1% osmium tetroxide solution, dehydrated in ethanol, and critical point dried by a Samdri-PVT-3D system (Tousimis, Maryland, USA). The specimens were coated with osmium tetroxide by osmium coater HPC-30W (Vacuum Device, Ibaraki, Japan). Electron micrographs were captured with an S-4800 field emission scanning electron microscope (Hitachi, Tokyo, Japan).

**Sperm motility analysis**. Cauda epididymal spermatozoa were dispersed in TYH medium[82] for analyzing sperm motility. At 10 and 120 min after the sperm incubation, velocity parameters were examined by using the CEROS II sperm analysis system (software version 1.5.2; Hamilton Thorne Biosciences, Massachusetts, USA). For analyzing flagellar bending patterns, sperm motility was videotaped at 200 frames per second using an Olympus BX-53 microscope equipped with a high-speed camera (HAS-L1, Ditect, Tokyo, Japan) as described previously[83]. Obtained frame images were analyzed using the sperm motion analyzing software (BohBohsoft, Tokyo, Japan)[84]. Single frames throughout one beating cycle were superimposed; in abnormal (bent or broken mid piece) spermatozoa with unclear beating cycle, 20 frames were superimposed.

**Sperm comet assay**. The alkaline comet assay was done using the CometAssay Kit (Trevigen). Specifically, spermatozoa were incubated in TYH medium for 30 min at 37 °C in an incubator with 5% $CO_2$. The spermatozoa were diluted with 50 μL PBS (final conc.: $1 \times 10^5$ spermatozoa/mL), mixed with molten low-melting-point-agarose (Trevigen), and then put on the comet slide. Sample slide were immersed in lysis solution (Trevigen) with 40 mM dithiothreitol at 4 °C for 20 min, and then Actinase (final conc.: 10 μg/mL) was added. After incubation for 2 h at room temperature, the slides were immersed in alkaline solution for 1 h at 4 °C. After electrophoresis for 40 min at 1 Volt/cm, the slides were stained by SYBR Gold (Thermo Fisher Scientific, Massachusetts, USA). The comet tail length of about 30 spermatozoa per slide was measured with ImageJ software.

**In vitro fertilization and embryo culture**. Mouse in vitro fertilization (IVF) was performed as described previously[85]. In brief, 2-month-old females were given an injection of 5 IU of pregnant mare serum gonadotropin (PMSG; ASKA Pharmaceutical Co., Tokyo, Japan), followed 48 h later by 5 IU of human chorionic gonadotropin (hCG; ASKA Pharmaceutical Co.) for superovulation. At 14 h after hCG injection, oocytes with cumulus cells (cumulus-oocyte complexes: COCs) were collected in 100 μL drops of TYH medium covered with paraffin oil. Cauda epididymal spermatozoa were collected from males and incubated in TYH medium for 2 h for capacitation. Capacitated spermatozoa were added to each drop containing COCs at a final concentration of $2 \times 10^5$ spermatozoa/mL. To remove cumulus cells, COCs were treated with hyaluronidase (1 mg/mL) for 5 min. At 8 h after incubation (post-insemination), the formation of pronuclei was observed under a Hoffman modulation contrast microscope. After IVF (Day 0), the embryos were cultured in KSOM medium up to Day 4 (from 1-cell to blastocyst stages) and observed under a phase contrast microscope. Blastocyst diameters were measured by ImageJ software.

**Immunostaining on Testis and epidymis sections**. Testes and epididymides were fixed in Bouin's solution for 2 h at room temperature, embedded in paraffin, and sectioned at 5 μm thickness. Slides were de-waxed, rehydrated, and heated in 10 mM sodium citrate buffer (pH 6.0). The slides were then blocked with 2.5% horse serum (Vector Laboratories, California, USA) for 30 min, incubated with primary antibodies for 1 h, washed with PBS, incubated with the secondary antibodies (ImmPRESS-HRP detection kit, Vector Laboratories) for 30 min, and washed with PBS, at room temperature. The primary antibodies are as follows: anti-STRA8 (1:500 dilution; ab49405, Abcam, Massachusetts, USA), anti-BrdU (1:500 dilution; ab6326, Abcam), anti-SOX9 (1:200 dilution; AB5535, Millipore, Massachusetts, USA), and anti-γH2AX (1:200 dilution; ab11174, Abcam). For colorimetric detection, slides were developed using a DAB substrate kit (Vector Laboratories), counterstained with Mayer's hematoxylin, dehydrated, and mounted with Permount (Fisher Scientific).

**Immunostaining on blastocysts**. Blastocysts were fixed in 4% paraformaldehyde for 40 min at 4 °C, washed with PBS, permeabilized with 0.5% Triton X-100 in PBS for 20 min, and washed with PBS. The blastocysts were then blocked with 2.5% goat serum for 45 min, incubated overnight with primary antibodies at 4 °C, washed with PBS, incubated with the secondary antibodies for 30 min at room temperature, and washed with PBS. The combinations of primary and secondary antibodies are as follows. Anti-OCT4 (1:200 dilution; PM048, MBL Life Science,

Nagoya, Japan) with goat anti-rabbit DyLight 549 (1:250 dilution, Thermo Fisher Scientific); and anti-CDX2 (1:150 dilution; B-MU392AUC, BioGenex, California, USA) with goat anti-mouse DyLight 488 (1:250 dilution, Thermo Fisher Scientific). Finally, blastocysts were whole-mounted with Antifade Mounting Medium with DAPI (H1200, Vector Laboratories) and observed under a confocal laser scanning microscope (C2+, Nikon, Japan). Z-stacks were collected at 0.4 μm intervals, and the maximum intensity projection images were generated from the z-stacks with total 10–12 μm thickness.

**Senescence-associated β-galactosidase staining**. Testes and epididymides were embedded in O.C.T. compound (Sakura Finetek, Tokyo, Japan) and frozen in liquid nitrogen. Frozen blocks were sectioned at 8 μm thickness, fixed in 2% paraformaldehyde (containing 0.2% glutaraldehyde) for 15 min at room temperature, washed with PBS, incubated in staining solution (containing 40 mM citric acid, 5 mM $K_4[Fe(CN)_6]3H_2O$, 5 mM $K_3[Fe(CN)_6]$, and 1 mg/mL X-gal)[39] for 20 h at 37 °C, and washed with PBS. Slides were counterstained with Nuclear Fast Red solution, dehydrated, and mounted with Permount (Fisher Scientific).

**RNA-Seq analysis**. Testes and epididymides were placed in TRIzol (Thermo Fisher Scientific), homogenized, and stored at −20 °C. Total RNAs were prepared according to the manufacturer's protocol. Libraries were prepared using a TruSeq Stranded mRNA Sample Prep Kit (Illumina, California, USA) according to the manufacturer's protocol. Whole-transcriptome sequencing was applied to the RNA samples using an Illumina HiSeq 2500 platform in a 75-base single-end mode. Sequenced reads were mapped to the mouse reference genome sequences (mm10) using TopHat version 2.0.13 in combination with Bowtie2 version 2.2.0 and SAMtools version 0.1.18. Normalized FPKM were calculated using Cuffnorm or Cuffdiff version 2.2.1 and each value lower than 0.1 was set to 0.1. Gene Ontology (GO) analysis was performed using Enrichr[86]. Heatmaps of differentially expressed genes were calculated with Subio software and visualized with Microsoft Excel.

**Statistics and reproducibility**. Statistical analysis was performed using GraphPad Prism 9. Data are represented as mean ± SD of three or more biological replicates. When comparing two groups, the $t$ test (one-tailed as indicated) or the $\chi^2$ test was employed. To compare three or more groups, Pearson's correlation or one-way ANOVA with the Tukey-Kramer *post hoc* test was used. To compare multiple experimental groups with a control group, one-way ANOVA with Dunnett's *post hoc* test was used. The sample sizes represent independent biological replicates. All experiments were performed at least three times.

**Reporting summary**. Further information on research design is available in the Nature Portfolio Reporting Summary linked to this article.

## Data availability

The source data behind the graphs are available in Supplementary Data 4. The obtained RNA-seq data have been deposited in the NCBI Gene Expression Omnibus (accession code GSE226150) (https://www.ncbi.nlm.nih.gov/geo/). All other data are available from the corresponding author (or other sources, as applicable) on reasonable request. Further inquiries can be directed to the corresponding author (atendo@g.ecc.u-tokyo.ac.jp).

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

## Acknowledgements

We thank Noriko Asai and Yoshimichi Miyashiro (ASKA Pharma Medical) for testicular testosterone measurement, Dr. Yoshitaka Fujihara, Dr. Masami Kanai-Azuma, Dr. Tatsuyuki Matsudaira, Dr. Koji Sugiura, and Yuki Tanaka for helpful discussion and technical assistance, and Dr. Julio M. Castaneda for critical reading of the manuscript. This work was supported by the Ministry of Education, Culture, Sports, Science and Technology (MEXT)/Japan Society for the Promotion of Science (JSPS) KAKENHI grants (JP19K06439 and JP23K05587 to T.E., and JP19H05750 and JP21H05033 to M.I.), Japan Agency for Medical Research and Development (AMED) grant JP21gm5010001 to M.I., and Takeda Science Foundation grants to M.I.

## Author contributions

T.E., E.H., and M.I. designed the research; T.E., K.K., T.M., C.E., M.O., S.K., Keisuke S., Kentaro S., M.K., and D.M. performed the research; T.E., D.O., Keisuke S., H.M., and Y.I. analyzed the data; and T.E. and M.I. wrote the paper.

## Competing interests

The authors declare no competing interests.
