## [Peer Review File · Communications Biology]

Reviewers' comments:

Reviewer #1 (Remarks to the Author):

Within the present manuscript entitled „Decreased sperm production in aged testes and functional sperm defects in aged epididymides lead to decreased fertility in male mice“, Endo et al. aimed at investigating the influence of ageing on male germ cell differentiation. Within this regard, the authors focused on assessing the overall organization of testis and epididymis as well as sperm functionality and male fertility in aged mice. As indicated by the title of the manuscript, the main findings of the manuscripts support an overall decrease of male fertility and sperm production with age. Intriguingly, these impairments are less dependent on mating behavior and testosterone levels, but rather due to defects in germ cell differentiation processes.

Although the manuscript contains some new experimental approaches (RNASeq of young vs old mice), almost no novel aspects were described in the present manuscript. It is well known that ageing has negative effects on male germ cell differentiation (quantitatively and qualitatively impaired spermatogenesis characterized by seminiferous tubule degeneration, germ cell depletion and morphological impairments of spermatozoa) in both men and mice - even in other species. The effects in experimental mice are summarized in multiple reviews e.g. by Carrageta et al. 2022 Reviews in Endocrine and Metabolic Disorders 23:1341-1360. Therefore, the authors need to clarify the novelty of the manuscript and clearly separate new findings from already well established knowledge.

Major comments:

- The authors claimed a decrease of epididymal spermatozoa in aged mice. The data included in the manuscript (epididymal sperm numbers) support the hypothesis that spermatogenesis is not only qualitatively but also quantitatively impaired. However, although sperm extracted from the cauda epididymidis provides a relative measure of the sperm count, superior methods are available that allow a more precise determination of sperm quantification. It is therefore recommended to assess the daily sperm production in addition to the epididymal sperm count.
- It needs to be elucidated why epididymal spermatozoa are decreased. Although the authors mentioned a decrease of type A spermatogonia it remains unclear why this is and if also other reasons may exist. It is recommended to assess whether the aged testis contains an increased number of apoptotic germ cells (e.g. by immunostaining against cleaved caspase 3 or TUNEL assay) and/ or whether differences in the cell differentiation processes exist (e.g. by KI67 or PCNA staining and subsequent stage-wise comparison of stages with increased rates of mitotic and meiotic divisions).
- The authors claimed that the effects seen in aged mice are independent of testosterone levels. However, 23-month-old mice have a substantial increase of senescence-associated β -galactosidase in the interstitial compartment, presumably Leydig cell clusters. At this stage, it is recommended to explain what an increase of SA- β -Gal would mean for the activity of cells.
- In general, the authors claimed that the spermatogenic defects and decreased male fertility is independent on testosterone levels and showed that testosterone levels were comparable between young and old mice. However, multiple studies have demonstrated in the past that the serum testosterone levels are decreased with age (e.g. Machida et al. Hormones and Behavior 1981; 15(3):238-245 or summarized by Carrageta et al. 2022 Reviews in Endocrine and Metabolic Disorders 23:1341-1360). Please explain this discrepancy among studies.
- Figure 6 shows morphology and motility of epididymal spermatozoa. The present data are rather superficial and would benefit from an in-depth analysis including ultrastructural assessment of morphological defects and in vitro capacitation assays or breeding experiments.
- The study lacks information about germ cell differentiation and sperm quality in older men. It is recommended to include respective literature into the introductory part or discussion.

Minor comments:

- Line 82: the phrase "by mating tests" does not fit into the remaining sentence. Please revise.
- It is recommended to revise the figure arrangement. In particular, figure 2a, b and c would better fit together with figure 1b (regarding the content of the shown diagrams)
- Figure 2d-j: It is recommended to increase the size of the histological images that would allow a better visibility of the image content (the diagrams next to the histology images can be slightly decreased in size).
- The plural of epididymis is epididymides. Please revise throughout the manuscript.

Reviewer #2 (Remarks to the Author):

In this study by Endo and colleagues, a detailed characterization of testicular and epididymal aging using mice as a model was conducted. Although several previous studies have added bits and pieces of information about the impacts of aging on normal spermatogenesis and fertility in mice, none have provided a comprehensive assessment and significant gaps in understanding still exist. Thus, this study, provides important new information for the field. The manuscript is well written, and experimentation is, for the most part, technically sound. However, there are a few major areas that I feel require further consideration and potentially further experimentation.

Major

I suggest reassessing the fecundity data presented for Figure 1d. There is no difference in pup numbers per litter between 2-4 months and 10-12 months of age. The significant change happens at 12-14 months of age compared to 2-12 months of age and then there is no further decline. This suggests that there is not necessarily a correlation of age with decline in fecundity. Rather, there is a trigger at 12-14 months of age that reduces fecundity to a stable level in advanced age. The authors should consider applying a Pearson's correlation or chi-square statistical analysis to the data.

Considering that several previous studies have measured a decline in serum testosterone in aging male animals, including mice, I am surprised by the data in Figure 1g showing no difference between 2 and 24 months of age. The data could be real or could reflect lack of sensitivity from the ELISA assay. I suggest that the authors consider a different means of measuring testosterone such as isotope dilution MS or LC-MS/MS which are more sensitive than ELISA. Also, the authors should consider measuring intratesticular testosterone rather than serum testosterone. Intratesticular testosterone is the dictator of androgen action on spermatogenesis and the level may not be accurately reflected by serum levels. Much of the data on a decline in quantitative and qualitative spermatogenesis could be explained by reduction in intratesticular testosterone levels, thus it is imperative that the authors' analysis is accurate to draw valid interpretations. At present, I struggle to agree with the conclusion that in vivo male fertility decreases with age regardless of testosterone levels.

Like the breeding data in Figure 1g, I suggest that the authors reanalyze the testis weight data in Figures 2b and 2c, epididymal weight data in Figures 3e and 3f, and IVF data in Figures 7a and 7b with Pearson's correlation or chi-square statistical tests rather than multiple comparisons ANOVA.

Lastly, I struggle to interpret the in vitro fertilization (IVF) and embryo culture data presented for Figure 7. As such, I cannot fully agree with the authors conclusions. First, fertilization rates presented in Figure 7a show that there is no difference for most of the aging process, until 24 months of age. Thus, I cannot agree with conclusions suggesting association of low IVF rates with aging. There is only one aged timepoint in which fertilization rates were different compared to younger age points. Second,

all of the embryo development data are comparisons between those generated from sperm of males at 2 and >24 months of age. Thus, a correlation between aging and outcomes from ARTs cannot be discerned. Rather, the data indicate that embryos derived from young adult and geriatric males are different. I suggest that the authors reconsider conclusions derived from these data about impacts of the aging process on IVF and embryo development rates.

Minor

Statements on in vitro aging of SSCs (lines 68-70) should be reconsidered. The two studies referenced in support of the statements used primary cultures derived from neonatal mice. The germ cells in the neonatal period are different compared to the spermatogonia present in an adult testis. Thus, yes, the transplantable cells in those cultures did age between 24-30 months of in vitro maintenance but the population is likely not reflective of adult spermatogonia. There are other studies that have investigated aging of primary SSC cultures generated from adult mice that should be considered (e.g. Helsel et al., 2017, Stem Cell Reports, PMID: 28392219).

We thank the reviewers for their careful readings and thoughtful comments for improving the manuscript. Revisions made in response to reviewers' comments are detailed below (in blue). In the revised manuscript, the changes made in sentences are shown in red.

Reviewer #1:

Within the present manuscript entitled „Decreased sperm production in aged testes and functional sperm defects in aged epididymides lead to decreased fertility in male mice”, Endo et al. aimed at investigating the influence of ageing on male germ cell differentiation. Within this regard, the authors focused on assessing the overall organization of testis and epididymis as well as sperm functionality and male fertility in aged mice. As indicated by the title of the manuscript, the main findings of the manuscripts support an overall decrease of male fertility and sperm production with age. Intriguingly, these impairments are less dependent on mating behavior and testosterone levels, but rather due to defects in germ cell differentiation processes.

Although the manuscript contains some new experimental approaches (RNASeq of young vs old mice), almost no novel aspects were described in the present manuscript. It is well known that ageing has negative effects on male germ cell differentiation (quantitatively and qualitatively impaired spermatogenesis characterized by seminiferous tubule degeneration, germ cell depletion and morphological impairments of spermatozoa) in both men and mice - even in other species. The effects in experimental mice are summarized in multiple reviews e.g. by Carrageta et al. 2022 *Reviews in Endocrine and Metabolic Disorders* 23:1341-1360. Therefore, the authors need to clarify the novelty of the manuscript and clearly separate new findings from already well established knowledge.

We appreciate the reviewer's careful reading of our manuscript. In response, we have revised throughout the manuscript to clarify the novelty of our findings and cited the literature (Carrageta et al., 2022) (p. 5 line 76 – p. 6 line 85; p. 18 lines 299 – 302). Moreover, we have incorporated additional data, new analysis, and additional display items (Fig. 1g, 4b-d, 5c, 6c, 8e, and Supplementary Fig. 6a-b), and modified the text and figures. We believe that these revisions enhance the quality and significance of our manuscript.

Major comments:

- The authors claimed a decrease of epididymal spermatozoa in aged mice. The data included in the manuscript (epididymal sperm numbers) support the hypothesis that spermatogenesis is not only qualitatively but also quantitatively impaired. However, although sperm extracted from the cauda epididymidis provides a relative measure of the sperm count, superior methods are available that allow a more precise determination of sperm quantification. It is therefore recommended to assess the daily sperm production in addition to the epididymal sperm count.

We have revised the Result section to explain that epididymal sperm count is a widely used and sensitive method of assessing sperm production; especially, when the sperm release (the end point of spermatogenesis) is impaired, epididymal sperm count is more sensitive than the testicular sperm count known as the daily sperm production (Lanning et al., 2002, *Toxicol Pathol*, 30, 507-20; Wang et al., 2003, *Curr Protoc Toxicol*, Chapter 16) (p. 9 lines 145 – 149).

- It needs to be elucidated why epididymal spermatozoa are decreased. Although the authors mentioned a decrease of type A spermatogonia it remains unclear why this is and if also other reasons may exist. It is recommended to assess whether the aged testis contains an increased number of apoptotic germ cells (e.g. by immunostaining against cleaved caspase 3 or TUNEL assay) and/ or whether differences in the cell differentiation processes exist (e.g. by KI67 or PCNA staining and subsequent stage-wise comparison of stages with increased rates of mitotic and meiotic divisions).

In response, we have counted apoptotic degenerating spermatogonia histologically because the histological assay is widely used to identify apoptotic degenerating cells in testes (de Rooij et al., 1999, *Biol Reprod*, 61, 842-7; Russell et al., 1990, Cache River Press) and because TUNEL assay in testes yields false positive staining (Dutta et al., *Toxicologic Pathology* 2012, 40: 667-674). The results, now shown in Fig. 4c-d, indicate that the number of degenerating spermatogonia was not significantly changed with age. We have also performed a stage-wise comparison of cell divisions in type A spermatogonia by BrdU assay, now shown in Fig. 4b. We found that BrdU-positive type A spermatogonia were decreased specifically in stages X-XII and I-III, when undifferentiated type A spermatogonia proliferate, and they were not changed in stages VII-IX, when type A spermatogonia undergo cell division for their differentiation toward spermatocytes.

We thus believe that, the decreased proliferation of type A spermatogonia, together with the resultant germ cell depletion (Fig. 2e-f) and sperm release failure (Fig. 2g-i), leads to a decreased number of epididymal spermatozoa. We have described these results in the text (p. 9 line 154 – p. 11 line 173; p. 18 line 310 – p. 19 line 322).

- The authors claimed that the effects seen in aged mice are independent of testosterone levels. However, 23-month-old mice have a substantial increase of senescence-associated β -galactosidase in the interstitial compartment, presumably Leydig cell clusters. At this stage, it is recommended to explain what an increase of SA- β -Gal would mean for the activity of cells.

We have revised the Discussion section (p. 20 lines 343 – 348) to explain that the interstitial compartment comprises blood vessels, T cells, and dendritic cells, all of which are known to undergo senescence and inflammation with age. Moreover, although we do not exclude the possibility that aged Leydig cells may have SA- β -Gal activity, functional decline of testosterone production has yet to occur at least at 24 months of age. We have also added the RNA-seq data, now shown in Fig. 5c, that key marker genes for testosterone production in Leydig cells were not decreased at 24 months of age, consistent with the testosterone levels.

- In general, the authors claimed that the spermatogenic defects and decreased male fertility is independent on testosterone levels and showed that testosterone levels were comparable between young and old mice. However, multiple studies have demonstrated in the past that the serum testosterone levels are decreased with age (e.g. Machida et al. *Hormones and Behavior* 1981; 15(3):238-245 or summarized by Carrageta et al. 2022 *Reviews in Endocrine and Metabolic Disorders* 23:1341-1360). Please explain this discrepancy among studies.

The difference between previous reports and our finding likely reflects differences in strains/genetic backgrounds of mice and/or housing environments. In particular, Machida

et al. (1981) used CD-1 (ICR) mice with an average lifespan of 15 months (450 days) in their condition and showed decreased serum testosterone levels after 15 months. Because we used C57BL/6J mice whose average lifespan is 29 months (Yuan et al., 2012, PNAS, 109, 8224-9), our finding that serum testosterone levels were not decreased until 24 months of age in C57BL/6J mice does not conflict with Machida et al.'s finding. We have revised the text (p. 18 lines 299 – 302) to clarify these experimental differences.

We have also measured intratesticular testosterone levels, now shown in Fig. 1g, and found that these levels were not significantly decreased at 24 months, which was consistent with serum testosterone levels. Please see our response below to Reviewer #2's comment.

- Figure 6 shows morphology and motility of epididymal spermatozoa. The present data are rather superficial and would benefit from an in-depth analysis including ultrastructural assessment of morphological defects and in vitro capacitation assays or breeding experiments.

As an in-depth analysis, we have performed an ultrastructural assessment by SEM, shown in Fig. 6c and Supplementary Fig. 6a-b, and found that aged spermatozoa had structurally abnormal flagella; the midpiece was abnormally curved next to the sperm head or broken at the annulus with a split mitochondrial alignment. Moreover, some aged spermatozoa had midpieces that lack mitochondria. These structural abnormalities were consistent with, and well explained, abnormal morphology (Fig. 6a-b), inflexible flagellar bending patterns (Fig. 6d), and decreased motility of spermatozoa (Fig. 6e-f). We have described these results in the text (p. 14 line 234 – p. 15 line 247).

- The study lacks information about germ cell differentiation and sperm quality in older men. It is recommended to include respective literature into the introductory part or discussion.

We have included the literature in the Discussion section and revised the text (p. 21 line 368 – p. 22 line 371).

Minor comments:

- Line 82: the phrase “by mating tests” does not fit into the remaining sentence. Please revise.

We have revised the text (p. 6 lines 88 – 90).

- It is recommended to revise the figure arrangement. In particular, figure 2a, b and c would better fit together with figure 1b (regarding the content of the shown diagrams)

We have moved Fig. 1b to Fig. 2b, and now it is displayed together with Fig. 2a-d (formerly Fig. 2a, b, and c).

- Figure 2d-j: It is recommended to increase the size of the histological images that would allow a better visibility of the image content (the diagrams next to the histology images can be slightly decreased in size).

We have increased the size of the histological images in Fig. 2d-j.

- The plural of epididymis is epididymides. Please revise throughout the manuscript.

We have revised throughout the manuscript.

Reviewer #2:

In this study by Endo and colleagues, a detailed characterization of testicular and epididymal aging using mice as a model was conducted. Although several previous studies have added bits and pieces of information about the impacts of aging on normal spermatogenesis and fertility in mice, none have provided a comprehensive assessment and significant gaps in understanding still exist. Thus, this study, provides important new information for the field. The manuscript is well written, and experimentation is, for the most part, technically sound.

We appreciate these comments on the significance of our findings and the quality of the presentation.

However, there are a few major areas that I feel require further consideration and potentially further experimentation.

Major

I suggest reassessing the fecundity data presented for Figure 1d. There is no difference in pup numbers per litter between 2-4 months and 10-12 months of age. The significant change happens at 12-14 months of age compared to 2-12 months of age and then there is no further decline. This suggests that there is not necessarily a correlation of age with decline in fecundity. Rather, there is a trigger at 12-14 months of age that reduces fecundity to a stable level in advanced age. The authors should consider applying a Pearson's correlation or chi-square statistical analysis to the data.

As suggested, we have reanalyzed the fecundity data presented for Fig. 1d (now Fig. 1c) with Pearson's correlation and found that the number of pups per litter showed a significant negative correlation with age from 2 to 24 months ($P = 0.0003$; $r = -0.45$). This statistical result indicates that the number of pups per litter gradually decreased until 24 months of age, although 12-14 months transiently showed a low average number (with a high statistical variability caused by one mouse). We have revised the text (p. 6 lines 90 – 92), Figure, and Figure legends.

Considering that several previous studies have measured a decline in serum testosterone in aging male animals, including mice, I am surprised by the data in Figure 1g showing no difference between 2 and 24 months of age. The data could be real or could reflect lack of sensitivity from the ELISA assay. I suggest that the authors consider a different means of measuring testosterone such as isotope dilution MS or LC-MS/MS which are more sensitive than ELISA. Also, the authors should consider measuring intratesticular testosterone rather than serum testosterone. Intratesticular testosterone is the dictator of androgen action on spermatogenesis and the level may not be accurately reflected by serum levels. Much of the data on a decline in quantitative and qualitative spermatogenesis could be explained by reduction in intratesticular testosterone levels, thus it is imperative that the authors' analysis is accurate to draw valid interpretations. At present, I struggle to agree with the conclusion that in vivo male fertility decreases with age regardless of testosterone levels.

We have measured intratesticular testosterone levels using LC-MS/MS, supported technically by experts on testosterone measurement (ASKA Pharma Medical, Fujisawa, Japan) (Shibata et al., 2017, *Prostate*, 77, 672-80; Yazawa et al., 2021, *Front Endocrinol*, 12, 1-10). As now shown in Fig. 1g and described in the Results section (p. 7 lines 103 – 104), we found that intratesticular testosterone levels were not significantly decreased at 24 months of age in our mouse model, which was consistent with serum testosterone levels. Indeed, the abnormalities observed in testis at 24 months, such as germ cell depletion, sperm release failure, and decreased proliferation of spermatogonia (now in Fig. 4b), are different from the reported testicular abnormalities in testosterone deficient models: arrests of meiosis and spermatid elongation (De Gendt et al., 2004, *PNAS*, 101, 1327-32; Holdcraft et al., 2004, *Development*, 131, 459-67) (see Discussion: p. 18 lines 303 – 307).

Like the breeding data in Figure 1g, I suggest that the authors reanalyze the testis weight data in Figures 2b and 2c, epididymal weight data in Figures 3e and 3f, and IVF data in Figures 7a and 7b with Pearson's correlation or chi-square statistical tests rather than multiple comparisons ANOVA.

As suggested, we have reanalyzed the testis weight data in Fig. 2b and 2c (now Fig. 2b and 2d), epididymal weight data in Fig. 3e and 3f, and IVF data in Fig. 7a and b, with Pearson's correlation and revised these Figures and Figure legends.

Lastly, I struggle to interpret the in vitro fertilization (IVF) and embryo culture data presented for Figure 7. As such, I cannot fully agree with the authors conclusions. First, fertilization rates presented in Figure 7a show that there is no difference for most of the aging process, until 24 months of age. Thus, I cannot agree with conclusions suggesting association of low IVF rates with aging. There is only one aged timepoint in which fertilization rates were different compared to younger age points. Second, all of the embryo development data are comparisons between those generated from sperm of males at 2 and >24 months of age. Thus, a correlation between aging and outcomes from ARTs cannot be discerned. Rather, the data indicate that embryos derived from young adult and geriatric males are different. I suggest that the authors reconsider conclusions derived from these data about impacts of the aging process on IVF and embryo development rates.

First, we now include the reanalyzed IVF data by Pearson's correlation, showing that IVF rates showed significant negative correlation with age from 2 to 24 months ($P = 0.0002$; $r = -0.58$) (Fig. 7a). Second, we believe that there are not much of difference between 24 and ≥ 24 (24-26) month data, based on the facts that average lifespan of C57BL/6J mice is 29 months (Yuan et al., 2012, *PNAS*, 109, 8224-9), and that 24-26 months of age is a widely used time point for ageing research (Novosadová et al., 2018, *Sci Rep*, 8, 11668; Limbad et al., 2022, *iScience*, 25, 103848; Alam et al., 2022, *Dis Model Mech*, 15, dmm048256). We have incorporated these citations in the text (p. 23 lines 383 – 385).

Minor

Statements on in vitro aging of SSCs (lines 68-70) should be reconsidered. The two studies referenced in support of the statements used primary cultures derived from neonatal mice. The germ cells in the neonatal period are different compared to the spermatogonia present in an adult testis. Thus, yes, the transplantable cells in those cultures did age between 24-30 months of in vitro maintenance but the population is likely not reflective of adult

spermatogonia. There are other studies that have investigated aging of primary SSC cultures generated from adult mice that should be considered (e.g. Hessel et al., 2017, Stem Cell Reports, PMID: 28392219).

We have cited the literature (Hessel et al., 2017) and revised the text (p. 5 lines 71 – 76).

REVIEWERS' COMMENTS:

Reviewer #1 (Remarks to the Author):

Within the present study, Endo et al have assessed age-related changes in the male fertility status incl. overall testicular and epididymal function and structure.

In the present revised version of the manuscript, the authors have thoroughly and comprehensively addressed all my comments and concerns (see first report) as well as performed requested experiments, and included further explanations.

Overall, the revised version of this comprehensive study is well-written and -structured and provides important information required to gain a better understanding about the influence of ageing on male germ cell differentiation and fertility.

Reviewer #2 (Remarks to the Author):

The authors have done a good job addressing my questions and concerns. I have no further reservations about the study and feel that it provides several solid pieces of new information for the field.